# Counter Turing Test (CT²): AI-Generated Text Detection is Not as Easy as You May Think – Introducing *AI Detectability Index*

**Megha Chakraborty**[1]    **S.M Towhidul Islam Tonmoy**[2]    **S M Mehedi Zaman**[2]
**Krish Sharma**[3]    **Niyar R Barman**[3]    **Chandan Gupta**[4]    **Shreya Gautam**[5]
**Tanay Kumar**[5]    **Vinija Jain**[6,7†]    **Aman Chadha**[6,7†]    **Amit P. Sheth**[1]    **Amitava Das**[1]

[1]AI Institute, University of South Carolina, USA    [2]IUT, Bangladesh    [3]NIT Silchar, India
[4]IIIT Delhi, India    [5]BITS Mesra, India    [6]Stanford University, USA    [7]Amazon AI, USA

## Abstract

With the rise of prolific ChatGPT, the risk and consequences of AI-generated text has increased alarmingly. This triggered a series of events, including an open letter (Marcus, 2023), signed by thousands of researchers and tech leaders in March 2023, demanding a six-month moratorium on the training of AI systems more sophisticated than GPT-4. To address the inevitable question of ownership attribution for AI-generated artifacts, the US Copyright Office (Copyright-Office, 2023) released a statement stating that "If a work's traditional elements of authorship were produced by a machine, the work lacks human authorship and the Office will not register it". Furthermore, both the US (White-House, 2023) and the EU (European-Parliament, 2023) governments have recently drafted their initial proposals regarding the regulatory framework for AI. Given this cynosural spotlight on generative AI, AI-generated text detection (AGTD) has emerged as a topic that has already received immediate attention in research, with some initial methods having been proposed, soon followed by emergence of techniques to bypass detection. This paper introduces the *Counter Turing Test (CT²)*, a benchmark consisting of techniques aiming to offer a comprehensive evaluation of the robustness of existing AGTD techniques. Our empirical findings unequivocally highlight the fragility of the proposed AGTD methods under scrutiny. Amidst the extensive deliberations on policy-making for regulating AI development, it is of utmost importance to assess the detectability of content generated by LLMs. Thus, to establish a quantifiable spectrum facilitating the evaluation and ranking of LLMs according to their detectability levels, we propose the *AI Detectability Index (ADI)*. We conduct a thorough examination of 15 contemporary LLMs, empirically demonstrating that larger LLMs tend to have a higher ADI, indicating they are less detectable compared to smaller LLMs. We firmly believe that ADI holds significant value as a tool for the wider NLP community, with the potential to serve as a rubric in AI-related policy-making.

## 1 Proposed AI-Generated Text Detection Techniques (AGTD) – A Review

Recently, six methods and their combinations have been proposed for AGTD: *(i) watermarking, (ii) perplexity estimation, (iii) burstiness estimation, (iv) negative log-likelihood curvature, (v) stylometric variation*, and *(vi) classifier-based approaches*. This paper focuses on critiquing their robustness and presents empirical evidence demonstrating their brittleness.

**Watermarking:** Watermarking AI-generated text, first proposed by Wiggers (2022), entails the incorporation of an imperceptible signal to establish the authorship of a specific text with a high degree of certainty. This approach is analogous to encryption and decryption. Kirchenbauer et al. (2023a) ($w_{v1}$) were the first to present operational watermarking models for LLMs, but their initial proposal faced criticism. Sadasivan et al. (2023) shared their initial studies suggesting that paraphrasing can efficiently eliminate watermarks. In a subsequent paper (Kirchenbauer et al., 2023b) ($w_{v2}$), the

---

†Work does not relate to position at Amazon.

authors put forth evidently more resilient watermarking techniques, asserting that paraphrasing does not significantly disrupt watermark signals in this iteration of their research. By conducting extensive experiments (detailed in Section 3), our study provides a thorough investigation of the de-watermarking techniques $w_{v1}$ and $w_{v2}$, demonstrating that the watermarked texts generated by both methods can be circumvented, albeit with a slight decrease in de-watermarking accuracy observed with $w_{v2}$. These results further strengthen our contention that text watermarking is fragile and lacks reliability for real-life applications.

**Perplexity Estimation:** The hypothesis related to perplexity-based AGTD methods is that humans exhibit significant variation in linguistic constraints, syntax, vocabulary, and other factors (aka *perplexity*) from one sentence to another. In contrast, LLMs display a higher degree of consistency in their linguistic style and structure. Employing this hypothesis, GPTZero (Tian, 2023) devised an AGTD tool that posited the overall perplexity human-generated text should surpass that of AI-generated text, as in the equation: $logp_{\Theta}(h_{text}) - logp_{\Theta}(AI_{text}) \geq 0$ (Appendix C). Furthermore, GPTZero assumes that the variations in perplexity across sentences would also be lower for AI-generated text. This phenomenon could potentially be quantified by estimating the entropy for sentence-wise perplexity, as depicted in the equation: $E_{perp} = logp_{\Theta}[\Sigma_{k=1}^n(|s_h^k - s_h^{k+1}|)] - logp_{\Theta}[\Sigma_{k=1}^n(|s_{AI}^k - s_{AI}^{k+1}|)] \geq 0$; where $s_h^k$ and $s_{AI}^k$ represent $k^{th}$ sentences of human and AI-written text respectively.

**Burstiness Estimation:** *Burstiness* refers to the patterns observed in word choice and vocabulary size. GPTZero (Tian, 2023) was the first to introduce burstiness estimation for AGTD. In this context, the hypothesis suggests that AI-generated text displays a higher frequency of clusters or bursts of similar words or phrases within shorter sections of the text. In contrast, humans exhibit a broader variation in their lexical choices, showcasing a more extensive range of vocabulary. Let $\sigma_\tau$ denote the

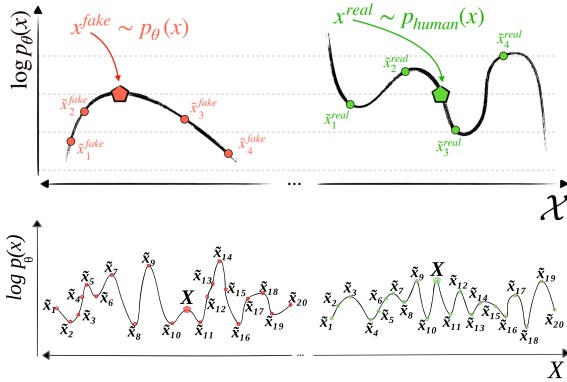

Figure 1: (Top) The negative log-curvature hypothesis proposed by Mitchell et al. (2023). According to their claim, any perturbations made to the AI-generated text should predominantly fall within a region of negative curvature. (Bottom) Our experiments using 15 LLMs with 20 perturbations indicate that the text generated by GPT 3.0 and variants do not align with this hypothesis. Moreover, for the other LLMs, the variance in the negative log-curvature was so minimal that it had to be disregarded as a reliable indication. ⬠ and ⬠ represent fake and real sample respectively, whereas 🔴 and 🟢 depict perturbed fake and real sample.

standard deviation of the language spans and $m_\tau$ the mean of the language spans. Burstiness ($b$) is calculated as $b = (\frac{\sigma_\tau/m_\tau - 1}{\sigma_\tau/m_\tau + 1})$ and is bounded within the interval [-1, 1]. Therefore the hypothesis is $b_H - b_{AI} \geq 0$, where $b_H$ is the mean burstiness of human writers and $b_{AI}$ is the mean burstiness of AI aka a particular LLM. Corpora with anti-bursty, periodic dispersions of switch points take on burstiness values closer to -1. In contrast, corpora with less predictable patterns of switching take on values closer to 1. It is worth noting that burstiness could also be calculated sentence-wise and/or text fragment-wise and then their entropy could be defined as: $E_{burst} = logp_{\beta}[\Sigma_{k=1}^n(|s_{AIb}^k - s_{AIb}^{k+1}|) - logp_{\beta}[\Sigma_{k=1}^n(|s_{hb}^k - s_{hb}^{k+1}|)]] \geq 0$. Nevertheless, our comprehensive experiments involving 15 LLMs indicate that this hypothesis does not consistently provide a discernible signal. Furthermore, recent LLMs like GPT-3.5/4, MPT (OpenAI, 2023a; Team, 2023) have demonstrated the utilization of a wide range of vocabulary, challenging the hypothesis. Section 4 discusses our experiments on perplexity and burstiness estimation.

**Negative Log-Curvature (NLC):** DetectGPT (Mitchell et al., 2023) introduced the concept of *Negative Log-Curvature (NLC)* to detect AI-generated text. The hypothesis is that text generated by the the model tends to lie in the negative curvature areas of the model's log probability, i.e. a text generated by a source LLM $p_\theta$ typically lies in the areas of negative curvature of the log probability function of $p_\theta$, unlike human-written text. In other words, we apply small perturbations to a passage $x \sim p_\theta$, producing $\tilde{x}$. Defining $P_\theta^{NLC}$ as the quantity $log\, p_\theta(x) - log\, p_\theta(\tilde{x})$, $P_\theta^{NLC}$ should be larger on average for AI-generated samples than human-written text (see an example in Table 1 and the visual intuition of the hypothesis in Fig. 1). Expressed mathematically: $P_{AI}^{NLC} - P_H^{NLC} \geq 0$. It is important to note that DetectGPT's findings were derived from text-snippet analysis, but there is potential to reevaluate this approach by examining smaller fragments, such as sentences. This would enable the calculation of averages or entropies, akin to how perplexity and burstiness are measured. Finally, the limited number of perturbation patterns per sentence in (Mitchell et al., 2023) affect the reliability of results (cf. Section 5 for details).

| Input Type | Sentence |
|---|---|
| Original | *This sentence is generated by an AI or human* |
| Perturbed | *This writing is created by an AI or person* |

Table 1: An example perturbation as proposed in DetectGPT (Mitchell et al., 2023).

**Stylometric variation:** Stylometry is dedicated to analyzing the linguistic style of text in order to differentiate between various writers. Kumarage et al. (2023) investigated the examination of stylistic features of AI-generated text in order to distinguish it from human-written text. The authors reported impressive results for text detection generated from RoBERTa. However, we observe limitations in applying such methods to newer advanced models (cf. Section 6).

**Classification-based approach:** This problem formulation involves training classifiers to differentiate between AI-written and human-written text,

and is relatively straightforward. OpenAI initially developed its own text classifier (OpenAI, 2023b), which reported an accuracy of only 26% on true positives. Due to its weaker performance among the proposed methods, we did not further investigate this strategy.

> **OUR CONTRIBUTIONS:** A Counter Turing Test ($CT^2$) and AI Detectability Index (ADI).
>
> ➠ Introducing the *Counter Turing Test (CT²)*, a benchmark consisting of techniques aiming to offer a comprehensive evaluation of the robustness of prevalent AGTD techniques.
>
> ➠ Empirically showing that the popular AGTD methods are brittle and relatively easy to circumvent.
>
> ➠ Introducing *AI Detectability Index (ADI)* as a measure for LLMs to infer whether their generations are detectable as AI-generated or not.
>
> ➠ Conducting a thorough examination of 15 contemporary LLMs to establish the aforementioned points.
>
> ➠ Both benchmarks – $CT^2$ and ADI – will be published as open-source leaderboards.
>
> ➠ Curated datasets will be made publicly available.

## 2 Design Choices for $CT^2$ and ADI Study

This section discusses our selected LLMs and elaborates on our data generation methods. More details in Appendix A.

### 2.1 LLMs: Rationale and Coverage

We chose a wide gamut of 15 LLMs that have exhibited exceptional results on a wide range of NLP tasks. They are: (i) GPT 4 (OpenAI, 2023a); (ii) GPT 3.5 (Chen et al., 2023); (iii) GPT 3 (Brown et al., 2020); (iv) GPT 2 (Radford et al., 2019); (v) MPT (Team, 2023); (vi) OPT (Zhang et al., 2022); (vii) LLaMA (Touvron et al., 2023); (viii) BLOOM (Scao et al., 2022); (ix) Alpaca (Maeng et al., 2017); (x) Vicuna (Zhu et al., 2023); (xi) Dolly (Wang et al., 2022); (xii) StableLM (Tow et al.); (xiii) XLNet (Yang et al., 2019); (xiv) T5 (Raffel et al., 2020); (xv) T0 (Sanh et al., 2021). Given that the field is ever-evolving, we admit that this process will never be complete but rather continue to expand. Hence, we plan to keep the

$CT^2$ benchmark leaderboard open to researchers, allowing for continuous updates and contributions.

## 2.2 Datasets: Generation and Statistics

To develop $CT^2$ and ADI, we utilize parallel data comprising both human-written and AI-generated text on the same topic. We select The New York Times (NYT) Twitter handle as our prompt source for the following reasons. Firstly, the handle comprises approximately 393K tweets that cover a variety of topics. For our work, we chose a subset of 100K tweets. Secondly, NYT is renowned for its reliability and credibility. The tweets from NYT exhibit a high level of word-craftsmanship by experienced journalists, devoid of grammatical mistakes. Thirdly, all the tweets from this source include URLs that lead to the corresponding human-written news articles. These tweets serve as prompts for the 15 LLMs, after eliminating hashtags and mentions during pre-processing. Appendix G offers the generated texts from 15 chosen LLMs when given the prompt *"AI generated text detection is not easy."*

## 3 De-Watermarking: Discovering its Ease and Efficiency

In the realm of philosophy, watermarking is typically regarded as a source-side activity. It is highly plausible that organizations engaged in the development and deployment of LLMs will progressively adopt this practice in the future. Additionally, regulatory mandates may necessitate the implementation of watermarking as an obligatory measure. The question that remains unanswered is the level of difficulty in circumventing watermarking, i.e., de-watermarking, when dealing with watermarked AI-generated text. In this section, we present our rigorous experiments that employ three methods capable of de-watermarking an AI-generated text that has been watermarked: (i) *spotting high entropy words and replacing them*, (ii) *paraphrasing*, (iii) *paraphrasing + replacing high-entropy words* Table 2 showcases an instance of de-watermarking utilizing two techniques for OPT as target LLM.

## 3.1 De-watermarking by Spotting and Replacing High Entropy Words ($DeW_1$)

The central concept behind the text watermarking proposed by Kirchenbauer et al. (2023a) is to identify high entropy words and replace them with alternative words that are contextually plausible. The replacement is chosen by an algorithm (analogous to an *encryption key*) known only to the LLM's creator. Hence, if watermarking has been implemented, it has specifically focused on those words. High entropy words are the content words in a linguistic construct. In contrast, low entropy words, such as function words, contribute to the linguistic structure and grammatical coherence of a given text. Replacing low entropy words can disrupt the quality of text generation. Appendix B provides more details on high entropy vs. low entropy words.

**Challenges of detecting high entropy words:** High entropy words aid in discerning ambiguity in LLM's as observed through the probability differences among predicted candidate words. While detecting high entropy words may seem technically feasible, there are two challenges in doing so: (i) many modern LLMs are not open-source. This restricts access to the LLM's probability distribution over the vocabulary; (ii) assuming a text snippet is AI-generated, in real-world scenarios, the specific LLM that generated it is challenging to determine unless explicitly stated. This lack of information makes it difficult to ascertain the origin and underlying generation process of a text.

**Spotting high-entropy words:** Closed-source LLMs conceal the log probabilities of generated text, thus rendering one of the most prevalent AGTD methods intractable. To address this, we utilize open-source LLMs to identify high-entropy words in a given text. As each LLM is trained on a distinct corpus, the specific high-entropy words identified may vary across different LLMs. To mitigate this, we adopt a comparative approach by employing multiple open-source LLMs.

**Replacing high-entropy words:** We can employ

| | | z-score | p-value |
|---|---|---|---|
| **Prompt** | *Will the next great writer be a robot?* | | |
| **Watermarked text** | I'm very skeptical that the next "great writer" is going to be a robot, or that they'll be much more effective at expressing the subtleties and depths of human thought than a human is. However, what is most interesting is the role that the Internet could play in bringing these "robot" writers into the public eye. If I could (and I'm very excited by this possibility), I would pay a small monthly fee to read well-written ... | 4.24 | $1.1 \times 10^{-5}$ |
| **De-Watermarked text by replacing high-entropy words** | I'm somewhat skeptical that the next "great writer" is going to be a robot, given that they'll be far more effective at grasping deeper subtleties and depths of philosophical thought than a robot is. However, what is particularly interesting is the role that the Internet may play in bringing new great writers into the public eye. If I did (and I'm extremely excited by this possibility), I would pay a hefty subscription fee to publish something ... | 1.76 (58.5% ↓) | 0.039 ($3.5 \times 10^5$% ↑) |
| **De-Watermarked text by para-phrasing** | I have serious doubts about the possibility of a robot becoming the next exceptional writer and surpassing humans in expressing the nuances and profoundness of human thoughts. Nevertheless, what fascinates me the most is the potential impact of the Internet in showcasing these "robot" writers to the general public. The idea of being able to pay a nominal monthly subscription fee to access impeccably written and carefully refined works truly thrills me... | -0.542 (112.8% ↓) | 0.706 ($6.4 \times 10^6$% ↑) |

Table 2: An illustration of de-watermarking by replacing high-entropy words and paraphrasing. p-value is the probability under the assumption of null hypothesis. The z-score indicates the normalized log probability of the original text obtained by subtracting the mean log probability of perturbed texts and dividing by the standard deviation of log probabilities of perturbed texts. DetectGPT (Mitchell et al., 2023) classifies text to be generated by GPT-2 if the z-score is greater than 4.

| Dewatermarking models → | albert-large-v2 | | | | bert-base-uncased | | | | distilroberta-base | | | | xlm-roberta-large | | | |
|---|---|---|---|---|---|---|---|---|---|---|---|---|---|---|---|---|
| | $DeW_1$ | | $DeW_2$ | | $DeW_1$ | | $DeW_2$ | | $DeW_1$ | | $DeW_2$ | | $DeW_1$ | | $DeW_2$ | |
| Masking models ↓ | $w_{v1}$ | $w_{v2}$ | $w_{v1}$ | $w_{v2}$ | $w_{v1}$ | $w_{v2}$ | $w_{v1}$ | $w_{v2}$ | $w_{v1}$ | $w_{v2}$ | $w_{v1}$ | $w_{v2}$ | $w_{v1}$ | $w_{v2}$ | $w_{v1}$ | $w_{v2}$ |
| albert-large-v2 | 51.8 | 68 | 99.5 | 99.3 | 47 | 71 | 99.87 | 99 | 75.8 | 70 | 99.55 | 98.45 | 62.5 | 59 | 99.09 | 96.56 |
| bert-base-uncased | 24.1 | 33 | 99.77 | 98.3 | 31 | 31 | 99.43 | 99.28 | 30.5 | 33 | 99.09 | 97.93 | 24 | 20 | 98.97 | 97.34 |
| distilroberta-base | 45.1 | 70 | 98.86 | 95.67 | 46.1 | 72 | 99.31 | 99.07 | 49.8 | 68 | 99.77 | 96.89 | 37.8 | 56 | 98.97 | 98.89 |
| xlm-roberta-large | 29.2 | 19 | 98.75 | 98.88 | 28.1 | 20 | 99.14 | 97.78 | 28.3 | 20 | 99.14 | 98.9 | 27.7 | 13 | 99.51 | 99.5 |

Table 3: The performance evaluation encompassed 16 combinations for de-watermarking **OPT** generated watermarked text. The accuracy scores for successfully de-watermarked text using the entropy-based word replacement technique are presented in the $DeW_1$ columns. It is worth highlighting that the accuracy scores in the $DeW_2$ columns reflect the application of automatic paraphrasing after entropy-based word replacement. The techniques proposed in Kirchenbauer et al. (2023a) are denoted as $w_{v1}$, while the techniques proposed in their subsequent work Kirchenbauer et al. (2023b) are represented as $w_{v2}$.

any LLM to replace the previously identified high-entropy words, resulting in a de-watermarked text. To achieve this, we tried various LLMs and found that BERT-based models are best performing to generate replacements for the masked text.

**Winning combination:** The results of experiments on detecting and replacing high entropy words are presented in Table 3 for OPT. The findings indicate that ALBERT (albert-large-v2) (Lan et al., 2020) and DistilRoBERTa (distilroberta-base) perform exceptionally well in identifying high entropy words in text generated by the OPT model for both versions, v1 and v2. On the other hand, DistilRoBERTa (distilroberta-base) (Sanh et al., 2019) and BERT (bert-base-uncased) (Devlin et al., 2019) demonstrate superior performance in substituting the high entropy words for versions v1 and v2 of the experiments. Therefore, the optimal combination for Kirchenbauer et al. (2023a) ($w_{v1}$) is (albert-large-v2,

distilroberta-base), achieving a 75% accuracy in removing watermarks, while (distilroberta-base, bert-base-uncased) performs best for (Kirchenbauer et al., 2023b) ($w_{v2}$), attaining 72% accuracy in de-watermarking. The results for the remaining 14 LLMs are reported in Appendix B.

**3.2 De-watermarking by Paraphrasing ($DeW_2$)**
We have used paraphrasing as yet another technique to remove watermarking from LLMs. Idea 1) Feed textual input to a paraphraser model such as Pegasus, T5, GPT-3.5 and evaluate watermarking for the paraphrased text. Idea 2) Replace the high entropy words, which are likely to be the watermarked tokens, and then paraphrase the text to ensure that we have eliminated the watermarks.

We perform a comprehensive analysis of both qualitative and quantitative aspects of automatic paraphrasing for the purpose of de-watermarking. We chose three SoTA paraphrase models: (a) Pegasus (Zhang et al., 2020), (b) T5 (Flan-t5-xxl

| LLMs | | Perplexity | | | | | Burstiness | | | | | NLC | | |
|---|---|---|---|---|---|---|---|---|---|---|---|---|---|---|
| | | Human | AI | Ent$_H$ | Ent$_{AI}$ | $\alpha$ | Human | AI | Ent$_H$ | Ent$_{AI}$ | $\alpha$ | Human | AI | $\alpha$ |
| OPT | $\mu$ | 46.839 | 43.495 | 4.276 | 3.777 | 0.519 | -0.3001 | 0.3645 | 6.119 | 5.890 | 0.5052 | 4.160 | 4.175 | 0.505 |
| | $\sigma$ | 68.541 | 65.178 | | | | 0.26164 | 0.3156 | | | | 0.336 | 0.654 | |
| GPT-2 | $\mu$ | 143.198 | 76.296 | 5.362 | 4.770 | 0.516 | -0.3001 | -0.2159 | 6.333 | 5.843 | 0.5006 | 3.436 | 3.778 | 0.507 |
| | $\sigma$ | 60.866 | 67.315 | | | | 0.26164 | 0.2947 | | | | 0.829 | 0.394 | |
| XLNet | $\mu$ | 106.776 | 104.091 | 8.378 | 9.712 | 0.532 | -0.2992 | -0.0153 | 6.380 | 4.563 | 0.4936 | 4.297 | 4.185 | 0.498 |
| | $\sigma$ | 57.091 | 62.152 | | | | 0.2416 | 0.0032 | | | | 0.338 | 0.535 | |

Table 4: Perplexity, burstiness, and NLC values for 3 LLMs across the ADI spectrum along with statistical measures.

| Paraphrasing Models | Acc. | |
|---|---|---|
| | $w_{v1}$ | $w_{v2}$ |
| Pegasus | 79.32 | 67.12 |
| T5-Large | 80.86 | 72.00 |
| GPT-3.5 | 90.32 | 70.35 |

Table 5: De-watermarking acc. of paraphrasing on OPT.

| Model | Coverage | Correctness | Diversity |
|---|---|---|---|
| Pegasus | 32.46 | 94.38% | 3.76 |
| T5 | 30.26 | 83.84% | 3.17 |
| GPT-3.5 | 35.51 | 88.16% | 7.72 |

Table 6: Experimental results of automatic paraphrasing models based on three factors: *(i) coverage, (ii) correctness and (iii) diversity*; GPT-3.5 (`gpt-3.5-turbo-0301`) can be seen as the most performant.

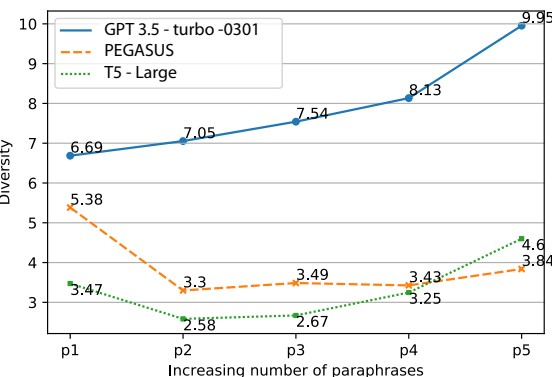

Figure 2: A higher diversity score depicts an increase in the number of generated paraphrases and linguistic variations in those generated paraphrases.

variant) (Chung et al., 2022), and (c) GPT-3.5 (`gpt-3.5-turbo-0301` variant) (Brown et al., 2020). We seek answers to the following questions: (i) *What is the accuracy of the paraphrases generated?* (ii) *How do they distort the original content?* (iii) *Are all the possible candidates generated by the paraphrase models successfully de-watermarked?* (iv) *Which paraphrase module has a greater impact on the de-watermarking process?* To address these questions, we evaluate the paraphrase modules based on three key dimensions: *(i) Coverage: number of considerable paraphrase generations, (ii) Correctness: correctness of the generations, (iii) Diversity: linguistic diversity in the generations*. Our experiments showed that GPT-3.5 (`gpt-3.5-turbo-0301` variant) is the most suitable paraphraser (Fig. 2). Please see details of experiments in Appendix B.3.

For a given text input, we generate multiple paraphrases using various SoTA models. In the process of choosing the appropriate paraphrase model based on a list of available models, the primary question we asked is how to make sure the generated paraphrases are rich in diversity while still being linguistically correct. We delineate the process followed to achieve this as follows. Let's say we have a claim $c$. We generate $n$ paraphrases us-

ing a paraphrasing model. This yields a set of $p_1^c$, ..., $p_n^c$. Next, we make pair-wise comparisons of these paraphrases with $c$, resulting in $c - p_1^c$, ..., and $c - p_n^c$. At this step, we identify the examples which are entailed, and only those are chosen. For the entailment task, we have utilized RoBERTa Large (Liu et al., 2019) – a SoTA model trained on the SNLI task (Bowman et al., 2015).

**Key Findings from De-Watermarking Experiments:** As shown in Table 3 and Table 5, our experiments provide empirical evidence suggesting that the watermarking applied to AI-generated text can be readily circumvented (cf. Appendix B).

## 4 Reliability of Perplexity and Burstiness as AGTD Signals

In this section, we extensively investigate the reliability of perplexity and burstiness as AGTD signals. Based on our empirical findings, it is evident that the text produced by newer LLMs is nearly

indistinguishable from human-written text from a statistical perspective.

The hypothesis assumes that AI-generated text displays a higher frequency of clusters or bursts of similar words or phrases within shorter sections of the text. In contrast, humans exhibit a broader variation in their lexical choices, showcasing a more extensive range of vocabulary. Moreover, sentence-wise human shows more variety in terms of length, and structure in comparison with AI-generated text. To measure this we have utilized entropy. The entropy $p_i log p_i$ of a random variable is the average level of *surprise*, or *uncertainty*.

## 4.1 Estimating Perplexity – Human vs. AI

**Perplexity** is a metric utilized for computing the probability of a given sequence of words in natural language. It is computed as $e^{-\frac{1}{N}\sum_{i=1}^{N}\log_2 p(w_i)}$, where $N$ represents the length of the word sequence, and $p(w_i)$ denotes the probability of the individual word $w_i$. As discussed previously, GPTZero (Tian, 2023) assumes that human-generated text exhibits more variations in both overall perplexity and sentence-wise perplexity as compared to AI-generated text. To evaluate the strength of this proposition, we compare text samples generated by 15 LLMs with corresponding human-generated text on the same topic. Our empirical findings indicate that larger LLMs, such as GPT-3+, closely resemble human-generated text and exhibit minimal distinctiveness. However, relatively smaller models such as XLNet, BLOOM, etc. are easily distinguishable from human-generated text. Fig. 3 demonstrates a side-by-side comparison of the overall perplexity of GPT4 and T5. We report results for 3 LLMs in Table 4 (cf. Table 22 in Appendix C for results over all 15 LLMs).

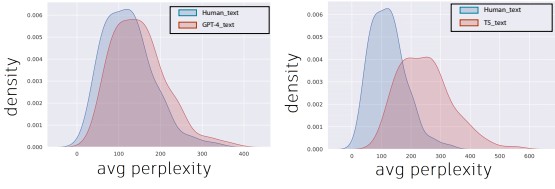

Figure 3: Perplexity estimation for GPT4/T5 (left/right).

## 4.2 Estimating Burstiness – Human vs. AI

In Section 1, we discussed the hypothesis that explores the contrasting **burstiness** patterns between human-written text and AI-generated text. Previous studies that have developed AGTD techniques based on burstiness include (Rychlỳ, 2011) and (Cummins, 2017). Table 4 shows that there is less distinction in the standard deviation of burstiness scores between AI-generated and human text for OPT. However, when it comes to XLNet, the difference becomes more pronounced. From several such examples, we infer that larger and more complex LLMs gave similar burstiness scores as humans. Hence, we conclude that as the size or complexity of the models increases, the deviation in burstiness scores diminishes. This, in turn, reinforces our claim that perplexity or burstiness estimations cannot be considered as reliable for AGTD (cf. Appendix C).

## 5 Negative Log-Curvature (NLC)

In Section 1, we discussed the NLC-based AGTD hypothesis (Mitchell et al., 2023). Our experimental results, depicted in Fig. 1, demonstrate that we are unable to corroborate the same NLC pattern for GPT4. To ensure the reliability of our experiments, we performed 20 perturbations per sentence. Fig. 1 (bottom) presents a comparative analysis of 20 perturbation patterns observed in 2000 samples of OPT-generated text and human-written text on the same topic. Regrettably, we do not see any discernible pattern. To fortify our conclusions, we compute the standard deviation, mean, and entropy, and conduct a statistical validity test using bootstrapping, which is more appropriate for non-Gaussian distributions (Kim, 2015; Boos and Brownie, 1989). Table 22 documents the results (cf. Appendix C). Based on our experimental results, we argue that NLC is not a robust method for AGTD.

## 6 Stylometric Variation

Stylometry analysis is a well-studied subject (Lagutina et al., 2019; Neal et al., 2018) where scholars have proposed a comprehensive range of lexical, syntactic, semantic, and structural characteristics for

the purpose of authorship attribution. Our investigation, which differs from the study conducted by Kumarage et al. (2023), represents the first attempt to explore the stylometric variations between human-written text and AI-generated text. Specifically, we assign 15 LLMs as distinct authors, whereas text composed by humans is presumed to originate from a hypothetical 16th author. Our task involves identifying stylometric variations among these 16 authors. After examining other alternatives put forth in previous studies such as (Tulchinskii et al., 2023), we encountered difficulties in drawing meaningful conclusions regarding the suitability of these methods for AGTD. Therefore, we focus our investigations on a specific approach that involves using perplexity (*as a syntactic feature*) and burstiness (*as a lexical choice feature*) as density functions to identify a specific LLM. By examining the range of values produced by these functions, we aim to pinpoint a specific LLM associated with a given text. Probability density such as $L_H^{plx} = \sum_{k=0}^{\infty} \left| Pr(S_{plx}^k) - \frac{\lambda_n^k e^{-\lambda_n}}{k!} \right|$ and $L_H^{brsty} = \sum_{k=0}^{\infty} \left| Pr(S_{brsty}^k) - \frac{\lambda_n^k e^{-\lambda_n}}{k!} \right|$ are calculated using Le Cam's lemma (Cam, 1986-2012), which gives the total variation distance between the sum of independent Bernoulli variables and a Poisson random variable with the same mean. Where $Pr(S_{plx}^k)$ is the perplexity and $Pr(S_{brsty}^k)$ is the brustiness of the of $k$th sentence respectively. In particular, it tells us that the sum is approximately Poisson in a specific sense (see more in Appendix E). Our experiment suggests stylistic feature estimation may not be very distinctive, with only broad ranges to group LLMs: (i) Detectable (80%+): T0 and T5, (ii) Hard to detect (70%+): XLNet, StableLM, and Dolly, and (iii) Impossible to detect (<50%): LLaMA, OPT, GPT, and variations.

Our experiment yielded intriguing results. Given that our stylometric analysis is solely based on density functions, we posed the question: what would happen if we learned the search density for one LLM and applied it to another LLM? To explore this, we generated a relational matrix, as depicted in Fig. 7. As previously described and illustrated

in Fig. 5, the LLMs can be classified into three groups: (i) easily detectable, (ii) hard to detect, and (iii) not detectable. Fig. 7 demonstrates that Le Cam's lemma learned for one LLM is only applicable to other LLMs within the same group. For instance, the lemma learned from GPT 4 can be successfully applied to GPT-3.5, OPT, and GPT-3, but not beyond that. Similarly, Vicuna, StableLM, and LLaMA form the second group. Fig. 4 offers a visual summary.

# 7 AI Detectability Index (ADI)

As new LLMs continue to emerge at an accelerated pace, the usability of prevailing AGTD techniques might not endure indefinitely. To align with the ever-changing landscape of LLMs, we introduce the AI Detectability Index (ADI), which identifies the discernable range for LLMs based on SoTA AGTD techniques. The hypothesis behind this proposal is that both LLMs and AGTD techniques' SoTA benchmarks can be regularly updated to adapt to the evolving landscape. Additionally, ADI serves as a litmus test to gauge whether contemporary LLMs have surpassed the ADI benchmark and are thereby rendering themselves impervious to detection, or whether new methods for AI-generated text detection will require the ADI standard to be reset and re-calibrated.

Among the various paradigms of AGTD, we select perplexity and burstiness as the foundation for quantifying the ADI. We contend that NLC is a derivative function of basic perplexity and burstiness, and if there are distinguishable patterns in NLC within AI-generated text, they should be well captured by perplexity and burstiness. We present a summary in Fig. 4 that illustrates the detectable and non-detectable sets of LLMs based on ADI scores obtained using stylometry and classification methods. It is evident that the detectable LLM set is relatively small for both paradigms, while the combination of perplexity and burstiness consistently provides a stable ADI spectrum. Furthermore, we argue that both stylistic features and classification are also derived functions of basic per-

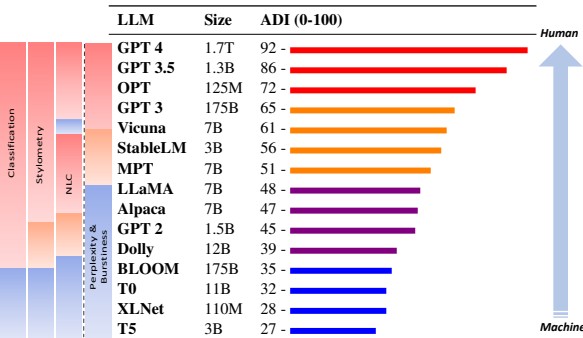

| LLM | Size | ADI (0-100) |
|-----|------|-------------|
| GPT 4 | 1.7T | 92 - |
| GPT 3.5 | 1.3B | 86 - |
| OPT | 125M | 72 - |
| GPT 3 | 175B | 65 - |
| Vicuna | 7B | 61 - |
| StableLM | 3B | 56 - |
| MPT | 7B | 51 - |
| LLaMA | 7B | 48 - |
| Alpaca | 7B | 47 - |
| GPT 2 | 1.5B | 45 - |
| Dolly | 12B | 39 - |
| BLOOM | 175B | 35 - |
| T0 | 11B | 32 - |
| XLNet | 110M | 28 - |
| T5 | 3B | 27 - |

Figure 4: ADI gamut for a diverse set of 15 LLMs.

plexity and burstiness. ADI serves to encapsulate the overall distinguishability between AI-written and human-written text, employing the formula:

$$ADI_x = \frac{100}{U \times 2} * [\sum_{x=1}^{U} \{\delta_1(x) * \frac{\left(P_t - L_H^{plx}\right)}{\left(1 - \mu_H^{plx}\right)}\} + \{\delta_2(x) * \frac{\left(B_t - L_H^{brsty}\right)}{\left(1 - \mu_H^{brsty}\right)}\}] \quad (1)$$

where, $P_t = \frac{1}{U} * \{\sum_{x=1}^{U} \left(log p_u^i - log p_u^{i+1}\right)\}$ and $B_t = \frac{1}{U} * \{\sum_{x=1}^{U} \left(log p_u^{i+(i+1)+(i+2)} - log p_u^{(i+3)+(i+4)+(i+5)}\right)\}$.

When confronted with a random input text, it is difficult to predict its resemblance to human-written text on the specific subject. Therefore, to calculate ADI we employ the mean perplexity ($\mu_H^{plx}$) and burstiness ($\mu_H^{brsty}$) derived from human-written text. Furthermore, to enhance the comparison between the current text and human text, Le Cam's lemma has been applied using pre-calculated values ($L_H^{plx}$ and $L_H^{brsty}$) as discussed in Section 6. To assess the overall contrast a summation has been used over all the 100K data points as depicted here by $U$. Lastly, comparative measures are needed to rank LLMs based on their detectability. This is achieved using multiplicative damping factors, $\delta_1(x)$ and $\delta_2(x)$, which are calculated based on $\mu \pm rank_x \times \sigma$. Initially, we calculate the ADI for all 15 LLMs, considering $\delta_1(x)$ and $\delta_2(x)$ as 0.5. With these initial ADIs, we obtain the mean ($\mu$) and standard deviation ($\sigma$), allowing us to recalculate the ADIs for all the LLMs. The resulting ADIs are then ranked and scaled providing a comparative spectrum as presented in Fig. 4. This scaling process is similar to Z-Score Normalization and/or Min-max normalization (Wikipedia,

2019). However, having damping factors is an easier option for exponential smoothing while we have a handful of data points. Finally, for better human readability ADI is scaled between $0 - 100$.

From the methods we considered, it is unlikely that any of them would be effective for models with high ADI, as shown by our experiments and results. As LLMs get more advanced, we assume that the current AGTD methods would become even more unreliable. With that in mind, ADI will remain a spectrum to judge which LLM is detectable and vs. which is not. Please refer to Appendix F for more discussion.

The ADI spectrum reveals the presence of three distinct groups. T0 and T5 are situated within the realm of *detectable range*, while XLNet, StableLM, Dolly, and Vicuna reside within the *difficult-to-detect range*. The remaining LLMs are deemed virtually impervious to detection through the utilization of prevailing SoTA AGTD techniques. It is conceivable that forthcoming advancements may lead to improved AGTD techniques and/or LLMs imbued with heightened human-like attributes that render them impossible to detect. Regardless of the unfolding future, ADI shall persist in serving the broader AI community and contribute to AI-related policy-making by identifying non-detectable LLMs that necessitate monitoring through policy control measures.

## 8 Conclusion

Our proposition is that SoTA AGTD techniques exhibit fragility. We provide empirical evidence to substantiate this argument by conducting experiments on 15 different LLMs. We proposed *AI Detectability Index (ADI)*, a quantifiable spectrum facilitating the evaluation and ranking of LLMs according to their detectability levels. The excitement and success of LLMs have resulted in their extensive proliferation, and this trend is anticipated to persist regardless of the future course they take. In light of this, the CT[2] benchmark and the *ADI* will continue to play a vital role in catering to the scientific community.

## 9 Ethical Considerations

Our experiments show the limitations of AGTD methods and how to bypass them. We develop ADI with the hope that it could be used for guiding further research and policies. However, it can be misused by bad actors for creating AI-generated text, particularly fake news, that cannot be distinguished from human-written text. We strongly advise against such use of our work.

## 10 Limitations

**Discussion:** On June 14th, 2023, the European Parliament successfully passed its version of the EU AI Act (European-Parliament, 2023). Subsequently, a team of researchers from the Stanford Institute for Human-Centered Artificial Intelligence (HAI) embarked on investigating the extent to which Foundation Model Providers comply with the EU AI Act. Their initial findings are presented in the publication by (Bommasani et al., 2023). In this study, the authors put forward a grading system consisting of 12 aspects for evaluating Language Models (LLMs). These aspects include *(i) data sources, (ii) data governance, (iii) copyrighted data, (iv) compute, (v) energy, (vi) capabilities & limitations, (vii) risk & mitigations, (viii) evaluation, (ix) testing, (x) machine-generated content, (xi) member states, and (xii) downstream documentation*. The overall grading of each LLM can be observed in Fig. 5. While this study is commendable, it appears to be inherently incomplete due to the ever-evolving nature of LLMs. Since all scores are assigned manually, any future changes will require a reassessment of this rubric, while ADI is auto-computable. Furthermore, we propose that ADI should be considered the most suitable metric for assessing risk and mitigations.

### 10.1 Addressing Opposing Views by Chakraborty et al. (2023)

It is important to note that a recent study (Chakraborty et al., 2023) contradicts our findings and claims otherwise. The study postulates that given enough sample points, whether the output was derived from a human vs an LLM is detectable, irrespective of the LLM used for AI-generated text. The sample size of this dataset is a function of the difference in the distribution of human text vs AI-text, with a smaller sample size enabling detection if the distributions show significant differences. However, the study does not provide empirical evidence or specify the required sample size, thus leaving the claim as a hypothesis at this stage.

Furthermore, the authors propose that employing techniques such as watermarking can change the distributions of AI text, making it more separable from human-text distribution and thus detectable. However, the main drawback of this argument is that given a single text snippet (say, an online article or a written essay), detecting whether it is AI-generated is not possible. Also, the proposed technique may not be cost-efficient compute-wise, especially as new LLMs emerge. However, the authors did not provide any empirical evidence to support this hypothesis.

**Limitations:** This paper delves into the discussion of six primary methods for AGTD and their potential combinations. These methods include (i) watermarking, (ii) perplexity estimation, (iii) burstiness estimation, (iv) negative log-likelihood curvature, (v) stylometric variation, and (vi) classifier-based approaches.

Our empirical research strongly indicates that the proposed methods are vulnerable to tampering or manipulation in various ways. We provide extensive empirical evidence to support this argument. However, it is important to acknowledge that there may still exist potential deficiencies in our experiments. In this section, we explore and discuss further avenues for investigation in order to address these potential shortcomings. In the subsequent paragraph, we outline the potential limitations associated with each of the methods we have previously investigated.

### 10.2 Watermarking

Although Kirchenbauer et al. (2023a) was the pioneering paper to introduce watermarking for AI-

# Grading Foundation Model Providers' Compliance with the Draft EU AI Act

Source: Stanford Center for Research on Foundation Models (CRFM), Institute for Human-Centered Artificial Intelligence (HAI)

| Draft AI Act Requirements | GPT-4 | Cohere Command | Stable Diffusion v2 | Claude | PaLM 2 | BLOOM | LLaMA | Jurassic-2 | Luminous | GPT-NeoX | Totals |
|---|---|---|---|---|---|---|---|---|---|---|---|
| Data sources | | | | | | | | | | | 22 |
| Data governance | | | | | | | | | | | 19 |
| Copyrighted data | | | | | | | | | | | 7 |
| Compute | | | | | | | | | | | 17 |
| Energy | | | | | | | | | | | 16 |
| Capabilities & limitations | | | | | | | | | | | 27 |
| Risks & mitigations | | | | | | | | | | | 16 |
| Evaluations | | | | | | | | | | | 15 |
| Testing | | | | | | | | | | | 10 |
| Machine-generated content | | | | | | | | | | | 21 |
| Member states | | | | | | | | | | | 9 |
| Downstream documentation | | | | | | | | | | | 24 |
| Totals | 25 / 48 | 23 / 48 | 22 / 48 | 7 / 48 | 27 / 48 | 36 / 48 | 21 / 48 | 8 / 48 | 5 / 48 | 29 / 48 | |

Figure 5: Grading of current LLMs as proposed by a report entitled *Do Foundation Model Providers Comply with the EU AI Act?* from Stanford University (Bommasani et al., 2023).

generated text, this research has encountered numerous criticisms since its inception. A major concern raised by several fellow researchers (Sadasivan et al., 2023) is that watermarking can be easily circumvented through machine-generated paraphrasing. In our experiment, we have presented two potential de-watermarking techniques. Subsequently, the same group of researchers published a follow-up paper (Kirchenbauer et al., 2023b) in which they asserted the development of a more advanced and robust watermarking technique. We assessed this claim as well and discovered that de-watermarking remains feasible. However, although the overall accuracy of de-watermarking has decreased, it still retains considerable strength. As the paper was published on June 9th, 2023, we will include the complete experiment details in the final version of our report.

In their work, Kirchenbauer et al. (2023b) put forward improved watermarking techniques by enhancing the hashing mechanism for selecting watermarking keys and introducing more effective watermark detection techniques. They conducted extensive testing on de-watermarking possibilities, considering both machine-generated paraphrasing and human paraphrasing, and observed dilution in the strength of the watermark, which aligns with their findings.

Although paraphrasing is a powerful technique for attacking watermark text, we argue that high-entropy-based word replacement offers a superior approach. When using high-entropy word replacements, it becomes exceedingly difficult for watermark detection modules to identify the newly generated text, even after paraphrasing. We will now elaborate on our rationale. In their work, Kirchenbauer et al. (2023b) identify content words such as nouns, verbs, adjectives, and adverbs as suitable candidates for replacement. However, any advanced techniques employed to select replacement watermark keys for these positions will result in high-entropy words. Consequently, these replacements will always remain detectable, regardless of the strength of the hashing mechanism.

## 10.3 Perplexity and Burstiness Estimation

Liang et al. (2023) and Chakraborty et al. (2023) among others have shown perplexity and burstiness are often not reliable indicators of human written text. The fallibility of these metrics become especially prominent in academic writing or text generated in a low-resource language. Our experiments have also pointed towards similar findings. Moreover, in our experiments, we computed perplexity and burstiness metrics both at the overall text level and the sentence level. It is also feasible to calculate perplexity at smaller fragment levels. Since each language model has a unique attention mechanism and span, these characteristics can potentially manifest in the generated text, making them detectable. However, determining the precise fragment size for a language model necessitates extensive experimentation, which we have not yet conducted.

## 10.4 Negative Log Curvature

Although we discussed earlier, it is crucial to re-emphasize the significant limitations of DetectGPT (Mitchell et al., 2023). One of its major limitations is that it relies on access to the log probabilities of the texts, which necessitates the use of a specific LLM. However, it is unlikely that we would know in advance which LLM was employed to generate a particular text, and the log-likelihood calculated by different LLMs for the same text would yield significantly different results. In reality, one would need to compare the results with all available LLMs in existence, which would require a computationally expensive brute-force search. In our experiments, we empirically demonstrate that the hypothesis of *log-probability #2 < log-probability #1* can be easily manipulated using simple [MASK]-based post-fixing techniques.

## 10.5 Stylometric Variation

In this experiment, we made a simplifying assumption that all the human-written text was authored by a single individual, which is certainly not re-

flective of reality. Furthermore, texts composed by different authors inevitably leave behind their unique traces and characteristics. Furthermore, a recent paper by Tulchinskii et al. (2023) introduced the concept of intrinsic dimensionality estimation, which can be described as a stylometric analysis. However, this paper is currently available only on arXiv and lacks an implemented solution. We are currently working on replicating the theory and evaluating the robustness of the approach.

## 10.6 Classifier-based Approaches

Numerous classifiers have been proposed in the literature (Zellers et al., 2020; Gehrmann et al., 2019; Solaiman et al., 2019). However, the majority of these classifiers are specifically created to identify instances generated by individual models. They achieve this by either utilizing the model itself (as demonstrated by Mitchell et al. (2023)) or by training on a dataset consisting of the generated samples from that particular model. For example, RoBERTa-Large-Detector developed by OpenAI (OpenAI, 2023b) is trained or fine-tuned specifically for binary classification tasks. These detectors are trained using datasets that consist of both human-generated and AI-generated texts. Consequently, their ability to effectively classify data from new models and unfamiliar domains is severely limited.

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

## Frequently Asked Questions (FAQs)

✳ **How do we envision ADI being used to influence LLM development, policy making, etc.?**

➠ The LLM has achieved the status of the *holy grail* in the field of AI. Its widespread adoption has been influenced by the success stories of ChatGPT, reaching various domains. As new LLMs continue to emerge regularly, there is a strong belief that future iterations will be even more powerful. Consequently, advanced AGTD techniques will be proposed to address these advancements. Regardless of the future landscape, the ADI will persist as a crucial tool for the scientific community and policymakers to assess the detectability of LLMs within their range.

✳ **For de-watermarking, Why do you use a brute force algorithm to choose a winning pair? Isn't it inefficient?**

➠ Our objective was to demonstrate the successful de-watermarking capability of a combination of open-source models. Currently, the combination of `albert-large-v2` and `distilroberta-base` has shown the most promising performance among all the LLMs. However, determining the most suitable combination for a text encountered in real-world scenarios poses a challenge. Exploring more efficient and scalable approaches to identify the optimal pair in such cases is an area that requires further investigation in future work.

✳ **For Stylometric analysis, the entire human-generated corpus was treated as if written by a single author. Won't that lead to noisy analysis?**

➠ Indeed, we made an easy presumption, but it opened up further possibilities.

✳ **Why did you compare only six methods?**

➠ We covered some of the most popular methods. It is possible but highly unlikely that there would be other contemporary methods which we did not try and are also very effective in AGTD.

✳ **Do you think your findings will generalize to other languages?**

➠ We have designed all the experiments and ADI in a way that is fairly applicable to any language. For example, de-watermarking techniques that we have discussed are based on "entropy" calculation, which is language agnostic. We have defined ADI primarily based on perplexity and burstiness, which could be applied for any language. Additionally, to expand the scope of our claim, we are already working on other languages, such as Spanish and Hindi, which we hope to publish soon.

# Appendix

This section provides supplementary material in the form of additional examples, implementation details, etc. to bolster the reader's understanding of the concepts presented in this work.

## A  LLM Selection Criteria

Beyond the primary criteria for choosing performant LLMs, our selection was meant to cover a wide gamut of LLMs that utilize a repertoire of recent techniques under the hood that have enabled their exceptional capabilities, namely: FlashAttention (Dao et al., 2022) for memory-efficient exact attention, Multi-Query Attention (Shazeer, 2019) for memory bandwidth efficiency, SwiGLU (Shazeer, 2020) as the activation function instead of ReLU (Agarap, 2019), ALiBi (Press et al., 2022) for larger context width, RMSNorm (Zhang and Sennrich, 2019) for per-normalization, RoPE (Su et al., 2021) to improve the expressivity of positional embeddings, etc.

## B  De-Watermarking

As also shown by Krishna et al. (2023), watermarked texts can be relatively easily de-watermarked. Even with the implementation of the newer, more robust watermarking scheme presented by Kirchenbauer et al. (2023b), we were still able to circumvent the watermarks to a significant extent. Here we discuss the methods in detail, concluding with Table 21 showing de-watermarking accuracies across 15 LLMs after paraphrasing.

### B.1  De-watermarking by spotting high entropy words and replacing them

The pivotal proposal made by the watermarking paper is to spot high entropy words and replace them with a random word from the vocabulary, so it is evident that if watermarking has been done, it has been done on those words.

**What are high entropy words?** High entropy words refer to words that are less predictable and occur less frequently in a corpus. These words have a higher degree of randomness and uncertainty and thus, pose a challenge for LLMs because they require a greater amount of training for accurate prediction. High entropy words can include domain-specific jargon or technical terms. Based on the observed patterns and frequencies of the training data, language models assign probabilities to words. Words with a high entropy tend to have lower probabilities because they are less common or have a more diverse contextual usage. These words are frequently uncommon or specialized terms, uncommon proper nouns, or words that are highly topic- or domain-specific. An example of such a high entropy word used in a sentence is as follows: "The adventurous child clambered up the gnarled tree, seeking the thrill of climbing to its lofty branches." In this sentence, the word "gnarled" is a high entropy word. It describes something that is twisted, rough, or knotted, typically referring to tree branches or old, weathered objects. In different language models, alternative words that might occur instead of "gnarled" could be "twisted," "knotty," or "weathered." These alternatives convey a similar meaning with more commonly used vocabulary. For instance, consider a masked input sentence: "Paris is the [MASK] of France." In this scenario, an LLM might predict candidate words with corresponding probabilities as follows: (i) "capital" [0.99], (ii) "city" [0.0], (iii) "metropolis" [0.0]. Here, the LLM demonstrates a high level of certainty regarding the word "capital" to fill the mask. Now, consider another sentence: "I saw a [MASK] last night." The LLM's predicted candidate words and their corresponding probabilities are: (i) "ghost" [0.096], (ii) "UFO" [0.083], (iii) "vampire" [0.045]. In this case, the LLM exhibits uncertainty in choosing the appropriate candidate word.

## B.2   Dewatemarking on 14 LLMs

Here we present performance evaluation of all the models' combination for the rest of the 14 LLMs. The "Pre" column shows the accuracy scores for the text that was successfully de-watermarked without any paraphrasing techniques. The "Post" column shows the accuracy scores for a text that was not successfully de-watermarked in the initial attempt but was able to be de-watermarked more successfully after paraphrasing methods were applied.

| Dewatermarking models → | albert-large-v2 | | | | bert-base-uncased | | | | distilroberta-base | | | | xlm-roberta-large | | | |
|---|---|---|---|---|---|---|---|---|---|---|---|---|---|---|---|---|
| Masking models ↓ | $DeW_1$ | | $DeW_2$ | | $DeW_1$ | | $DeW_2$ | | $DeW_1$ | | $DeW_2$ | | $DeW_1$ | | $DeW_2$ | |
| | $w_{v1}$ | $w_{v2}$ | $w_{v1}$ | $w_{v2}$ | $w_{v1}$ | $w_{v2}$ | $w_{v1}$ | $w_{v2}$ | $w_{v1}$ | $w_{v2}$ | $w_{v1}$ | $w_{v2}$ | $w_{v1}$ | $w_{v2}$ | $w_{v1}$ | $w_{v2}$ |
| albert-large-v2 | 91.9 | 96.5 | 93 | 98 | 93.3 | 96.5 | 95 | 98.3 | 91.1 | 93 | 92.5 | 96 | 87.8 | 91.9 | 90 | 94.3 |
| bert-base-uncased | 67.8 | 79.1 | 92.3 | 94.5 | 66.7 | 86 | 89.2 | 96.5 | 61.1 | 81.4 | 86.2 | 93.7 | 60 | 77.9 | 87.8 | 97.5 |
| distilroberta-base | 80 | 88.4 | 95.2 | 98 | 75.6 | 89.5 | 94 | 99.5 | 82.2 | 90.7 | 97.5 | 99.4 | 72.2 | 88.4 | 93.2 | 97 |
| xlm-roberta-large | 50 | 76.7 | 82 | 85.5 | 47.8 | 77.9 | 76.3 | 90 | 48.9 | 70.9 | 89 | 90 | 40 | 77.9 | 87.6 | 95.5 |

Table 7: Performance evaluation of 16 combinations of 4 masking-based models for de-watermarking **LLaMA** generated watermarked text.

| Dewatermarking models → | albert-large-v2 | | | | bert-base-uncased | | | | distilroberta-base | | | | xlm-roberta-large | | | |
|---|---|---|---|---|---|---|---|---|---|---|---|---|---|---|---|---|
| Masking models ↓ | $DeW_1$ | | $DeW_2$ | | $DeW_1$ | | $DeW_2$ | | $DeW_1$ | | $DeW_2$ | | $DeW_1$ | | $DeW_2$ | |
| | $w_{v1}$ | $w_{v2}$ | $w_{v1}$ | $w_{v2}$ | $w_{v1}$ | $w_{v2}$ | $w_{v1}$ | $w_{v2}$ | $w_{v1}$ | $w_{v2}$ | $w_{v1}$ | $w_{v2}$ | $w_{v1}$ | $w_{v2}$ | $w_{v1}$ | $w_{v2}$ |
| albert-large-v2 | 93.8 | 95.9 | 98.7 | 99.7 | 92.5 | 94.6 | 95 | 99 | 91.2 | 94.6 | 96.7 | 98 | 93.8 | 95.9 | 99 | 99.5 |
| bert-base-uncased | 66.2 | 78.4 | 88 | 90 | 73.8 | 79.7 | 96 | 98.5 | 70 | 85.1 | 90 | 95 | 67.5 | 78.4 | 90 | 97 |
| distilroberta-base | 90 | 90.5 | 92.5 | 96 | 86.3 | 91.9 | 90 | 98 | 87.5 | 91.9 | 97.5 | 99 | 86.3 | 89.2 | 96.5 | 95 |
| xlm-roberta-large | 53.7 | 85.1 | 78.9 | 90 | 53.7 | 82.4 | 79.5 | 91.5 | 53.7 | 87.8 | 84 | 95 | 45 | 81.1 | 69.8 | 82 |

Table 8: Performance evaluation of 16 combinations of 4 masking-based models for de-watermarking **Alpaca** generated watermarked text.

| Dewatermarking models → | albert-large-v2 | | | | bert-base-uncased | | | | distilroberta-base | | | | xlm-roberta-large | | | |
|---|---|---|---|---|---|---|---|---|---|---|---|---|---|---|---|---|
| Masking models ↓ | $DeW_1$ | | $DeW_2$ | | $DeW_1$ | | $DeW_2$ | | $DeW_1$ | | $DeW_2$ | | $DeW_1$ | | $DeW_2$ | |
| | $w_{v1}$ | $w_{v2}$ | $w_{v1}$ | $w_{v2}$ | $w_{v1}$ | $w_{v2}$ | $w_{v1}$ | $w_{v2}$ | $w_{v1}$ | $w_{v2}$ | $w_{v1}$ | $w_{v2}$ | $w_{v1}$ | $w_{v2}$ | $w_{v1}$ | $w_{v2}$ |
| albert-large-v2 | 64.6 | 73.7 | 90 | 98 | 76.8 | 56.6 | 97.5 | 88 | 78.8 | 82.8 | 95.5 | 95.5 | 76.8 | 84.8 | 95.5 | 95.5 |
| bert-base-uncased | 18.2 | 57.6 | 52.9 | 78.4 | 23.2 | 36.4 | 50 | 60.5 | 36.4 | 51.5 | 55.5 | 95 | 31.3 | 50.5 | 83.5 | 90 |
| distilroberta-base | 64.6 | 70.7 | 90 | 97 | 61.6 | 50.5 | 85.5 | 80 | 62.6 | 77.8 | 70.9 | 90 | 62.6 | 64.6 | 89 | 87 |
| xlm-roberta-large | 36.4 | 58.6 | 56.6 | 77.7 | 26.3 | 34.3 | 52.9 | 60 | 34.3 | 51.5 | 63.4 | 80.9 | 30.3 | 50.5 | 50 | 80.5 |

Table 9: Performance evaluation of 16 combinations of 4 masking-based models for de-watermarking **BLOOM** generated watermarked text.

| Dewatermarking models → | albert-large-v2 | | | | bert-base-uncased | | | | distilroberta-base | | | | xlm-roberta-large | | | |
|---|---|---|---|---|---|---|---|---|---|---|---|---|---|---|---|---|
| Masking models ↓ | $DeW_1$ | | $DeW_2$ | | $DeW_1$ | | $DeW_2$ | | $DeW_1$ | | $DeW_2$ | | $DeW_1$ | | $DeW_2$ | |
| | $w_{v1}$ | $w_{v2}$ | $w_{v1}$ | $w_{v2}$ | $w_{v1}$ | $w_{v2}$ | $w_{v1}$ | $w_{v2}$ | $w_{v1}$ | $w_{v2}$ | $w_{v1}$ | $w_{v2}$ | $w_{v1}$ | $w_{v2}$ | $w_{v1}$ | $w_{v2}$ |
| albert-large-v2 | 62.8 | 98 | 78 | 99.1 | 56.4 | 96 | 90 | 97 | 56.4 | 95 | 91 | 96 | 51.1 | 93 | 94.5 | 95 |
| bert-base-uncased | 28.7 | 77 | 99 | 100 | 26.6 | 70 | 94.6 | 96.7 | 26.6 | 74 | 90.5 | 96.2 | 24.5 | 77 | 88.5 | 95 |
| distilroberta-base | 48.9 | 91 | 84.5 | 100 | 40.4 | 89 | 70.5 | 92 | 51.1 | 92 | 90 | 95 | 38.3 | 91 | 75 | 94 |
| xlm-roberta-large | 29.8 | 77 | 50 | 83 | 24.5 | 46.5 | 86 | 95.5 | 27.7 | 76 | 50 | 76 | 26.6 | 72 | 54.5 | 96.4 |

Table 10: Performance evaluation of 16 combinations of 4 masking-based models for de-watermarking **StableLM** generated watermarked text.

| Dewatermarking models → | albert-large-v2 | | | | bert-base-uncased | | | | distilroberta-base | | | | xlm-roberta-large | | | |
|---|---|---|---|---|---|---|---|---|---|---|---|---|---|---|---|---|
| | $DeW_1$ | | $DeW_2$ | | $DeW_1$ | | $DeW_2$ | | $DeW_1$ | | $DeW_2$ | | $DeW_1$ | | $DeW_2$ | |
| Masking models ↓ | $w_{v1}$ | $w_{v2}$ | $w_{v1}$ | $w_{v2}$ | $w_{v1}$ | $w_{v2}$ | $w_{v1}$ | $w_{v2}$ | $w_{v1}$ | $w_{v2}$ | $w_{v1}$ | $w_{v2}$ | $w_{v1}$ | $w_{v2}$ | $w_{v1}$ | $w_{v2}$ |
| albert-large-v2 | 61.1 | 90.7 | 90 | 99 | 63.3 | 94.2 | 89 | 95 | 62.2 | 94.2 | 90.5 | 99 | 60 | 86 | 90 | 95.5 |
| bert-base-uncased | 41.1 | 67.4 | 80 | 88.6 | 41.1 | 67.4 | 80 | 88.6 | 46.7 | 69.8 | 84.5 | 90 | 40 | 61.6 | 80.5 | 91.5 |
| distilroberta-base | 51.1 | 80.2 | 80 | 98 | 56.7 | 83.7 | 78 | 88.9 | 53.3 | 87.2 | 83.9 | 98 | 51.1 | 76.7 | 85.5 | 90 |
| xlm-roberta-large | 33.3 | 56.9 | 58 | 76.5 | 40 | 55.8 | 80 | 97.6 | 42.2 | 52.3 | 87 | 89 | 44.4 | 47.7 | 80.3 | 89 |

Table 11: Performance evaluation of 16 combinations of 4 masking-based models for de-watermarking **Dolly** generated watermarked text.

| Dewatermarking models → | albert-large-v2 | | | | bert-base-uncased | | | | distilroberta-base | | | | xlm-roberta-large | | | |
|---|---|---|---|---|---|---|---|---|---|---|---|---|---|---|---|---|
| | $DeW_1$ | | $DeW_2$ | | $DeW_1$ | | $DeW_2$ | | $DeW_1$ | | $DeW_2$ | | $DeW_1$ | | $DeW_2$ | |
| Masking models ↓ | $w_{v1}$ | $w_{v2}$ | $w_{v1}$ | $w_{v2}$ | $w_{v1}$ | $w_{v2}$ | $w_{v1}$ | $w_{v2}$ | $w_{v1}$ | $w_{v2}$ | $w_{v1}$ | $w_{v2}$ | $w_{v1}$ | $w_{v2}$ | $w_{v1}$ | $w_{v2}$ |
| albert-large-v2 | 75 | 97 | 95 | 99.7 | 76.9 | 96 | 92.3 | 98.5 | 75 | 98 | 96.6 | 98.5 | 69.2 | 97.5 | 89 | 99.5 |
| bert-base-uncased | 61.5 | 90 | 87 | 94.5 | 65.4 | 89 | 90 | 92 | 57.7 | 92.5 | 80 | 95 | 65.4 | 94 | 85.5 | 94 |
| distilroberta-base | 71.2 | 99 | 89 | 99 | 76.9 | 94 | 85 | 96 | 75 | 92 | 90.4 | 99 | 65.4 | 97 | 90 | 98.5 |
| xlm-roberta-large | 59.6 | 97.5 | 90.7 | 97 | 63.5 | 97 | 87 | 99 | 53.8 | 97 | 85.3 | 98 | 55.8 | 98 | 80 | 98.2 |

Table 12: Performance evaluation of 16 combinations of 4 masking-based models for de-watermarking **T5** generated watermarked text.

| Dewatermarking models → | albert-large-v2 | | | | bert-base-uncased | | | | distilroberta-base | | | | xlm-roberta-large | | | |
|---|---|---|---|---|---|---|---|---|---|---|---|---|---|---|---|---|
| | $DeW_1$ | | $DeW_2$ | | $DeW_1$ | | $DeW_2$ | | $DeW_1$ | | $DeW_2$ | | $DeW_1$ | | $DeW_2$ | |
| Masking models ↓ | $w_{v1}$ | $w_{v2}$ | $w_{v1}$ | $w_{v2}$ | $w_{v1}$ | $w_{v2}$ | $w_{v1}$ | $w_{v2}$ | $w_{v1}$ | $w_{v2}$ | $w_{v1}$ | $w_{v2}$ | $w_{v1}$ | $w_{v2}$ | $w_{v1}$ | $w_{v2}$ |
| albert-large-v2 | 86.5 | 95 | 99 | 99.3 | 78.8 | 94.4 | 90 | 98.3 | 78.8 | 97 | 99 | 99.7 | 76.9 | 94.4 | 89.6 | 96.5 |
| bert-base-uncased | 71.2 | 88.9 | 90.2 | 99.7 | 71.2 | 94.4 | 86.2 | 95 | 73.1 | 88.9 | 90 | 95.8 | 67.3 | 94.4 | 76 | 96.8 |
| distilroberta-base | 69.2 | 94.4 | 79.5 | 96 | 75 | 95.6 | 95 | 97.3 | 82.7 | 94.4 | 96.5 | 98 | 71.2 | 94.4 | 91.2 | 98 |
| xlm-roberta-large | 57.7 | 94.4 | 70.9 | 95.7 | 65.4 | 94.4 | 85 | 96 | 59.6 | 88.9 | 80.5 | 90 | 61.5 | 88.9 | 90 | 98.5 |

Table 13: Performance evaluation of 16 combinations of 4 masking-based models for de-watermarking **Vicuna** generated watermarked text.

| Dewatermarking models → | albert-large-v2 | | | | bert-base-uncased | | | | distilroberta-base | | | | xlm-roberta-large | | | |
|---|---|---|---|---|---|---|---|---|---|---|---|---|---|---|---|---|
| | $DeW_1$ | | $DeW_2$ | | $DeW_1$ | | $DeW_2$ | | $DeW_1$ | | $DeW_2$ | | $DeW_1$ | | $DeW_2$ | |
| Masking models ↓ | $w_{v1}$ | $w_{v2}$ | $w_{v1}$ | $w_{v2}$ | $w_{v1}$ | $w_{v2}$ | $w_{v1}$ | $w_{v2}$ | $w_{v1}$ | $w_{v2}$ | $w_{v1}$ | $w_{v2}$ | $w_{v1}$ | $w_{v2}$ | $w_{v1}$ | $w_{v2}$ |
| albert-large-v2 | 80 | 86 | 86 | 88 | 81.4 | 87 | 90.8 | 87.5 | 80 | 86 | 94 | 88 | 67.1 | 76 | 96 | 79 |
| bert-base-uncased | 62.9 | 73 | 67.7 | 78 | 67.1 | 77 | 85.6 | 87 | 60 | 71 | 80.5 | 91.8 | 51.4 | 65 | 95 | 97.4 |
| distilroberta-base | 62.9 | 73 | 92.6 | 99 | 60 | 72 | 95 | 78 | 65.7 | 76 | 95 | 81 | 61.4 | 73 | 80.9 | 76 |
| xlm-roberta-large | 58.6 | 71 | 65 | 78 | 58.6 | 71 | 70.5 | 74 | 52.9 | 67 | 60.3 | 69 | 47.1 | 63 | 80 | 82 |

Table 14: Performance evaluation of 16 combinations of 4 masking-based models for de-watermarking **T0** generated watermarked text.

| Dewatermarking models → | albert-large-v2 | | | | bert-base-uncased | | | | distilroberta-base | | | | xlm-roberta-large | | | |
|---|---|---|---|---|---|---|---|---|---|---|---|---|---|---|---|---|
| | $DeW_1$ | | $DeW_2$ | | $DeW_1$ | | $DeW_2$ | | $DeW_1$ | | $DeW_2$ | | $DeW_1$ | | $DeW_2$ | |
| Masking models ↓ | $w_{v1}$ | $w_{v2}$ | $w_{v1}$ | $w_{v2}$ | $w_{v1}$ | $w_{v2}$ | $w_{v1}$ | $w_{v2}$ | $w_{v1}$ | $w_{v2}$ | $w_{v1}$ | $w_{v2}$ | $w_{v1}$ | $w_{v2}$ | $w_{v1}$ | $w_{v2}$ |
| albert-large-v2 | 87 | 87 | 92.3 | 92.3 | 92 | 92 | 93.5 | 93.5 | 88 | 88 | 93.3 | 93.3 | 93 | 93 | 95 | 95.7 |
| bert-base-uncased | 67 | 67 | 97 | 97 | 68 | 68 | 84.4 | 84.4 | 60 | 56.9 | 88.2 | 83.7 | 63 | 63 | 86.5 | 86.5 |
| distilroberta-base | 85 | 85 | 99 | 100 | 85 | 85 | 86.7 | 86.7 | 86 | 86 | 87.5 | 87.7 | 85 | 85 | 88.3 | 88.3 |
| xlm-roberta-large | 70 | 70 | 93.3 | 93.3 | 67 | 67 | 84.8 | 84.8 | 61 | 61 | 79.5 | 79.5 | 67 | 67 | 87.9 | 87.9 |

Table 15: Performance evaluation of 16 combinations of 4 masking-based models for de-watermarking **XLNet** generated watermarked text.

| Dewatermarking models → | albert-large-v2 | | | | bert-base-uncased | | | | distilroberta-base | | | | xlm-roberta-large | | | |
|---|---|---|---|---|---|---|---|---|---|---|---|---|---|---|---|---|
| Masking models ↓ | $DeW_1$ | | $DeW_2$ | | $DeW_1$ | | $DeW_2$ | | $DeW_1$ | | $DeW_2$ | | $DeW_1$ | | $DeW_2$ | |
| | $w_{v1}$ | $w_{v2}$ | $w_{v1}$ | $w_{v2}$ | $w_{v1}$ | $w_{v2}$ | $w_{v1}$ | $w_{v2}$ | $w_{v1}$ | $w_{v2}$ | $w_{v1}$ | $w_{v2}$ | $w_{v1}$ | $w_{v2}$ | $w_{v1}$ | $w_{v2}$ |
| albert-large-v2 | 97 | 91 | 99.2 | 99.5 | 98 | 88 | 99 | 98.5 | 95 | 88 | 99.8 | 90 | 97.5 | 88 | 99.8 | 93 |
| bert-base-uncased | 99 | 77 | 99.5 | 99.5 | 91 | 73 | 99.5 | 83 | 92 | 77 | 99.5 | 87 | 80 | 75 | 98.2 | 87.7 |
| distilroberta-base | 98 | 86 | 99 | 99 | 90 | 86 | 99 | 95 | 93 | 89 | 98 | 95.5 | 97 | 85 | 98.7 | 95.2 |
| xlm-roberta-large | 95 | 70 | 99.8 | 85 | 91 | 66 | 99.8 | 82 | 99 | 66 | 99.8 | 79.5 | 94 | 63 | 98 | 76 |

Table 16: Performance evaluation of 16 combinations of 4 masking-based models for de-watermarking **MPT** generated watermarked text.

| Dewatermarking models → | albert-large-v2 | | | | bert-base-uncased | | | | distilroberta-base | | | | xlm-roberta-large | | | |
|---|---|---|---|---|---|---|---|---|---|---|---|---|---|---|---|---|
| Masking models ↓ | $DeW_1$ | | $DeW_2$ | | $DeW_1$ | | $DeW_2$ | | $DeW_1$ | | $DeW_2$ | | $DeW_1$ | | $DeW_2$ | |
| | $w_{v1}$ | $w_{v2}$ | $w_{v1}$ | $w_{v2}$ | $w_{v1}$ | $w_{v2}$ | $w_{v1}$ | $w_{v2}$ | $w_{v1}$ | $w_{v2}$ | $w_{v1}$ | $w_{v2}$ | $w_{v1}$ | $w_{v2}$ | $w_{v1}$ | $w_{v2}$ |
| albert-large-v2 | 75 | 97 | 95 | 99.7 | 89 | 73.4 | 64.5 | 96 | 82.3 | 80 | 90 | 88.7 | 65 | 71 | 89.2 | 93.9 |
| bert-base-uncased | 52 | 69 | 70 | 89 | 78 | 80.5 | 87.9 | 96 | 85 | 89 | 98 | 99.7 | 76.2 | 80.3 | 90.5 | 92.3 |
| distilroberta-base | 75 | 80.9 | 80 | 81 | 61 | 61 | 78.9 | 78.9 | 64 | 74.2 | 74.5 | 90 | 87.8 | 90 | 87.8 | 90 |
| xlm-roberta-large | 67 | 68.9 | 77.5 | 82.4 | 80 | 90 | 84 | 99.7 | 52 | 54.5 | 67 | 70.5 | 34 | 56.6 | 50.4 | 60 |

Table 17: Performance evaluation of 16 combinations of 4 masking-based models for de-watermarking **GPT2** generated watermarked text.

| Dewatermarking models → | albert-large-v2 | | | | bert-base-uncased | | | | distilroberta-base | | | | xlm-roberta-large | | | |
|---|---|---|---|---|---|---|---|---|---|---|---|---|---|---|---|---|
| Masking models ↓ | $DeW_1$ | | $DeW_2$ | | $DeW_1$ | | $DeW_2$ | | $DeW_1$ | | $DeW_2$ | | $DeW_1$ | | $DeW_2$ | |
| | $w_{v1}$ | $w_{v2}$ | $w_{v1}$ | $w_{v2}$ | $w_{v1}$ | $w_{v2}$ | $w_{v1}$ | $w_{v2}$ | $w_{v1}$ | $w_{v2}$ | $w_{v1}$ | $w_{v2}$ | $w_{v1}$ | $w_{v2}$ | $w_{v1}$ | $w_{v2}$ |
| albert-large-v2 | 75 | 97 | 95 | 99.7 | 76.9 | 96 | 92.3 | 98.5 | 75 | 98 | 96.6 | 98.5 | 69.2 | 97.5 | 89 | 99.5 |
| bert-base-uncased | 61.5 | 90 | 87 | 94.5 | 65.4 | 89 | 90 | 92 | 57.7 | 92.5 | 80 | 95 | 57.7 | 94 | 85.5 | 94 |
| distilroberta-base | 71.2 | 99 | 89 | 99 | 76.9 | 94 | 85 | 96 | 75 | 92 | 90.4 | 99 | 65.4 | 97 | 90 | 98.5 |
| xlm-roberta-large | 59.6 | 97.5 | 90.7 | 97 | 63.5 | 97 | 87 | 99 | 53.8 | 97 | 85.3 | 98 | 55.8 | 98 | 80 | 98.2 |

Table 18: Performance evaluation of 16 combinations of 4 masking-based models for de-watermarking **GPT3** generated watermarked text.

| Dewatermarking models → | albert-large-v2 | | | | bert-base-uncased | | | | distilroberta-base | | | | xlm-roberta-large | | | |
|---|---|---|---|---|---|---|---|---|---|---|---|---|---|---|---|---|
| Masking models ↓ | $DeW_1$ | | $DeW_2$ | | $DeW_1$ | | $DeW_2$ | | $DeW_1$ | | $DeW_2$ | | $DeW_1$ | | $DeW_2$ | |
| | $w_{v1}$ | $w_{v2}$ | $w_{v1}$ | $w_{v2}$ | $w_{v1}$ | $w_{v2}$ | $w_{v1}$ | $w_{v2}$ | $w_{v1}$ | $w_{v2}$ | $w_{v1}$ | $w_{v2}$ | $w_{v1}$ | $w_{v2}$ | $w_{v1}$ | $w_{v2}$ |
| albert-large-v2 | 62.2 | 90.7 | 90 | 99 | 64.5 | 94.2 | 89 | 95 | 62.2 | 94.2 | 92.5 | 99 | 60 | 86 | 90 | 95.5 |
| bert-base-uncased | 41.1 | 67.4 | 80 | 88.6 | 41.1 | 67.4 | 80 | 88.6 | 46.7 | 69.8 | 84.5 | 90 | 46 | 61.6 | 80.5 | 91.5 |
| distilroberta-base | 51.1 | 80.2 | 80 | 98 | 56.7 | 83.7 | 78 | 88.9 | 53.3 | 87.2 | 85.9 | 98 | 51.1 | 76.7 | 85.5 | 90 |
| xlm-roberta-large | 33.3 | 56.9 | 59 | 76.5 | 40 | 56.8 | 80 | 97.6 | 42.2 | 52.3 | 87 | 89 | 47.4 | 47.7 | 80.3 | 89 |

Table 19: Performance evaluation of 16 combinations of 4 masking-based models for de-watermarking **GPT3.5** generated watermarked text.

| Dewatermarking models → | albert-large-v2 | | | | bert-base-uncased | | | | distilroberta-base | | | | xlm-roberta-large | | | |
|---|---|---|---|---|---|---|---|---|---|---|---|---|---|---|---|---|
| Masking models ↓ | $DeW_1$ | | $DeW_2$ | | $DeW_1$ | | $DeW_2$ | | $DeW_1$ | | $DeW_2$ | | $DeW_1$ | | $DeW_2$ | |
| | $w_{v1}$ | $w_{v2}$ | $w_{v1}$ | $w_{v2}$ | $w_{v1}$ | $w_{v2}$ | $w_{v1}$ | $w_{v2}$ | $w_{v1}$ | $w_{v2}$ | $w_{v1}$ | $w_{v2}$ | $w_{v1}$ | $w_{v2}$ | $w_{v1}$ | $w_{v2}$ |
| albert-large-v2 | 66 | 72 | 90 | 92.5 | 62 | 63 | 89 | 89.6 | 68 | 72.4 | 88.9 | 91.3 | 76 | 70 | 76 | 80.9 |
| bert-base-uncased | 58.6 | 71 | 66 | 78 | 58.6 | 71 | 70.5 | 74 | 52.9 | 67 | 60.3 | 69 | 47.1 | 63 | 83 | 82 |
| distilroberta-base | 95 | 76 | 99.8 | 85 | 91 | 66 | 98.8 | 82 | 99 | 66 | 99.8 | 80.5 | 94 | 63 | 98 | 76 |
| xlm-roberta-large | 67 | 89.3 | 90 | 99.7 | 80 | 83 | 83 | 89.2 | 89 | 95 | 90.5 | 98.7 | 76 | 85 | 86.5 | 96 |

Table 20: Performance evaluation of 16 combinations of 4 masking-based models for de-watermarking **GPT4** generated watermarked text.

## B.3 De-watermarking by paraphrasing

A recent paper (Krishna et al., 2023) talks about the DIPPER paraphrasing technique and how it can easily bypass the watermarking technique. However, their de-watermarking strategy can reduce the detection accuracy of the watermark detector tool to a certain extent. It can't fully de-watermark all the texts.

Another paper (Sadasivan et al., 2023) also uses the DIPPER paraphrasing technique but a slightly modified version in which they use parallel paraphrasing of multiple sentences. However, in this paper, they came up with how to bypass the paraphrasing technique so that even after paraphrasing, the detector can tell if the text is in fact AI-generated. This bypassing technique was named Retrieval and it uses the semantic sequence to detect AI-generated text even after paraphrasing (Krishna et al., 2023).

Both these papers also talk about the negative log-likelihood and perplexity score and they have tried on GPT and OPT models.

Based on empirical observations, we concluded that GPT-3.5 outperformed all the other models. To offer transparency around our experiment process, we detail the aforementioned evaluation dimensions as follows.

**Coverage - number of considerable paraphrase generations:** We intend to generate up to 5 paraphrases per given claim. Given all the generated claims, we perform a minimum edit distance (MED) (Wagner and Fischer, 1974) - units are words instead of alphabets). If MED is greater than $\pm 2$ for any given paraphrase candidate (for e.g., $c - p_1^c$) with the claim, then we further consider that paraphrase, otherwise discarded. We evaluated all three models based on this setup that what model is generating the maximum number of considerable paraphrases.

**Correctness - correctness in those generations:** After the first level of filtration we have performed pairwise entailment and kept only those paraphrase candidates, are marked as entailed by the (Liu et al., 2019) (Roberta Large), SoTA trained on SNLI (Bowman et al., 2015).

**Diversity - linguistic diversity in those generations:** We were interested in choosing that model can produce linguistically more diverse paraphrases. Therefore we are interested in the dissimilarities check between generated paraphrase claims. For e.g., $c - p_n^c$, $p_1^c - p_n^c$, $p_2^c - p_n^c$, $\ldots$, $p_{n-1}^c - p_n^c$ and repeat this process for all the other paraphrases and average out the dissimilarity score. There is no such metric to measure dissimilarity, therefore we use the inverse of the BLEU score (Papineni et al., 2002). This gives us an understanding of how linguistic diversity is produced by a given model. Based on these experiments, we found that `gpt-3.5-turbo-0301` performed the best. The results of the experiment are reported in the following table. Furthermore, we were more interested to choose a model that

| Models | Paraphrase | | | | | |
| | GPT-3.5-Turbo | | Pegasus | | Flan-T5-XXL | |
| | $w_{v1}$ | $w_{v2}$ | $w_{v1}$ | $w_{v2}$ | $w_{v1}$ | $w_{v2}$ |
|---|---|---|---|---|---|---|
| GPT 4 | 88% | 73% | 79% | 69% | 78% | 68% |
| GPT 3.5 | 89% | 72% | 78% | 68% | 79% | 69% |
| OPT | 90% | 70% | 79% | 67% | 80% | 72% |
| GPT 3 | 91% | 70% | 82% | 68% | 81% | 73% |
| Vicuna | 93% | 74% | 85% | 70% | 82% | 75% |
| StableLM | 95.0% | 98.0% | 96.4% | 87.0% | 83.0% | 42.5% |
| MPT | 96.0% | 99.0% | 88.5% | 90.1% | 85.0% | 68.7% |
| LLaMA | 95.0% | 98.7% | 89.3% | 99.1% | 98.0% | 98.9% |
| Alpaca | 95.0% | 99.0% | 95.5% | 99.0% | 70.5% | 66.7% |
| GPT 2 | 70.3% | 91.0% | 89.0% | 79.5% | 68.0% | 99.0% |
| Dolly | 98.0% | 96.0% | 95.6% | 91.6% | 98.0% | 70.9% |
| BLOOM | 97.0% | 97.0% | 87.2% | 92.9% | 85.5% | 76.8% |
| T0 | 98.0% | 99.0% | 96.8% | 96.0% | 83.9% | 80.0% |
| XLNet | 91.3% | 88.0% | 98.3% | 89.7% | 63.3% | 63.0% |
| T5 | 97.7% | 99.1% | 99.0% | 98.4% | 98.9% | 99.2% |

Table 21: A summary of the effectiveness of the three paraphrasing methods - a) Pegasus (Zhang et al., 2020), (b) Flan-t5-xxl (Chung et al., 2022), and (c) GPT-3.5 (`gpt-3.5-turbo-0301` variant) (Ye et al., 2023) for de-watermarking.

can maximize the linguistic variations, and `gpt-3.5-turbo-0301` performs on this parameter of choice as well. A plot on diversity vs. all the chosen models is reported in Fig. 2.

Table 21 provides a summary of the effectiveness of the three paraphrasing methods for de-watermarking. Among them, the GPT3.5 based method demonstrated the highest performance. Additionally, it is worth noting that the de-watermarking accuracy for $w_{v2}$, the watermarking technique proposed in (Kirchenbauer et al., 2023b), showed a slight decrease compared to $w_{v1}$, the watermarking technique proposed in (Kirchenbauer et al., 2023a).

## C Perplexity and Burstiness Estimation

We have conducted an analysis to determine the perplexity and burstiness of an LLM, as well as calculate sentence-wise entropy. In order to evaluate the statistical significance of our findings, we employed the bootstrap method. Results of these experiments on all 15 models are reported in Table 22.

**Brief on bootstrap method:** Bootstrapping is a statistical procedure that resamples a single dataset to create many simulated samples, illustrated in Fig. 6. This process allows for the calculation of standard errors, confidence intervals, and hypothesis testing. A bootstrapping approach is an extremely useful alternative to the traditional method of hypothesis testing as it is fairly simple and it mitigates some of the pitfalls encountered within the traditional approach. As with the traditional approach, a sample of size $n$ is drawn from the population within the bootstrapping approach. Let us call this sample $S$. Then, rather than using theory to determine all possible estimates, the sampling distribution is created by resampling observations with replacement from $S$, m times, with each resampled set having $n$ observations. Now, if sampled appropriately,

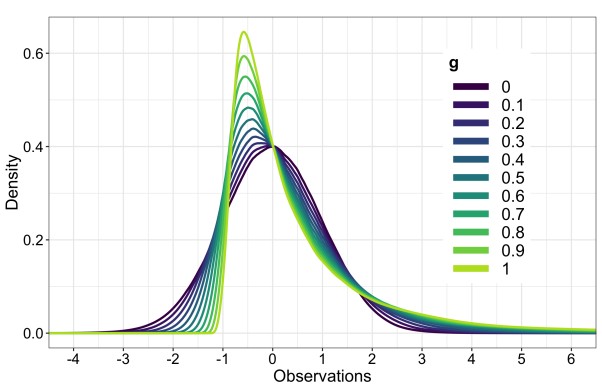

Figure 6: An illustration of Bootstrapping method – how it creates simulated samples.

$S$ should be representative of the population. Therefore, by resampling $S$ m times with replacement, it would be as if m samples were drawn from the original population, and the estimates derived would be representative of the theoretical distribution under the traditional approach. It must be noted that increasing the number of resamples, m, will not increase the amount of information in the data. That is, resampling the original set $100,000$ times is not more useful than only resampling it $1,000$ times. The amount of information within the set is dependent on the sample size, $n$, which will remain constant throughout each resample. The benefit of more resamples, then, is to derive a better estimate of the sampling distribution. The traditional procedure requires one to have a test statistic that satisfies particular assumptions in order to achieve valid results, and this is largely dependent on the experimental design. The traditional approach also uses theory to tell what the sampling distribution should look like, but the results fall apart if the assumptions of the theory are not met. The bootstrapping method, on the other hand, takes the original sample data and then resamples it to create many [simulated] samples. This approach does not rely on the theory since the sampling distribution can simply be observed, and one does not have to worry about any assumptions. This technique allows for accurate estimates of statistics, which is crucial when using data to make decisions.

## C.1 Reliability of Perplexity, Burstiness and NLC as AGT Signals for all LLMs

Here we present the complete table showing results after performing experiments on Perplexity estimation (Section 4.1), Burstiness estimation (Section 4.2) and NLC (Section 5) over all 15 LLMs.

| LLMs | | Perplexity | | | | | Burstiness | | | | | NLC | | |
|---|---|---|---|---|---|---|---|---|---|---|---|---|---|---|
| | | Human | AI | $Ent_H$ | $Ent_{AI}$ | $\alpha$ | Human | AI | $Ent_H$ | $Ent_{AI}$ | $\alpha$ | Human | AI | $\alpha$ |
| GPT 4 (OpenAI, 2023a) | $\mu$ | 38.073 | 35.465 | 4.222 | 3.881 | 0.492 | -0.4010 | 0.3920 | 6.152 | 5.893 | 0.5004 | 2.123 | 1.966 | 0.503 |
| | $\sigma$ | 86.411 | 80.836 | | | | 0.26164 | 0.3421 | | | | 0.535 | 0.934 | |
| GPT 3.5 (Chen et al., 2023) | $\mu$ | 43.198 | 39.897 | 3.423 | 3.195 | 0.505 | -0.2798 | 0.5509 | 6.144 | 5.923 | 0.5029 | 4.492 | 4.302 | 0.504 |
| | $\sigma$ | 46.866 | 42.341 | | | | 0.2966 | 0.3387 | | | | 0.332 | 0.514 | |
| OPT (Zhang et al., 2022) | $\mu$ | 46.839 | 43.495 | 4.276 | 3.777 | 0.519 | -0.3001 | 0.3645 | 6.119 | 5.890 | 0.5052 | 4.160 | 4.175 | 0.505 |
| | $\sigma$ | 68.541 | 65.178 | | | | 0.26164 | 0.3156 | | | | 0.336 | 0.654 | |
| GPT 3 (Brown et al., 2020) | $\mu$ | 48.839 | 46.980 | 4.205 | 3.933 | 0.515 | -0.3001 | 0.3171 | 6.119 | 5.880 | 0.5104 | 4.160 | 4.302 | 0.503 |
| | $\sigma$ | 82.541 | 79.224 | | | | 0.26164 | 0.2420 | | | | 0.336 | 0.465 | |
| Vicuna (Zhu et al., 2023) | $\mu$ | 51.839 | 50.728 | 4.276 | 3.676 | 0.511 | -0.3001 | 0.3122 | 6.119 | 5.763 | 0.5009 | 4.160 | 4.491 | 0.507 |
| | $\sigma$ | 58.541 | 50.740 | | | | 0.26164 | 0.3066 | | | | 0.336 | 0.427 | |
| StableLM (Tow et al.) | $\mu$ | 62.839 | 56.558 | 4.205 | 3.564 | 0.506 | -0.3001 | 0.1213 | 6.901 | 5.841 | 0.4945 | 4.160 | 4.386 | 0.499 |
| | $\sigma$ | 58.104 | 50.002 | | | | 0.26164 | 0.3434 | | | | 0.336 | 0.551 | |
| MPT (Team, 2023) | $\mu$ | 78.839 | 72.495 | 4.263 | 3.406 | 0.505 | -0.3001 | 0.1958 | 6.213 | 5.571 | 0.5041 | 4.160 | 4.260 | 0.486 |
| | $\sigma$ | 76.541 | 66.634 | | | | 0.26164 | 0.3834 | | | | 0.336 | 0.626 | |
| LLaMA (Touvron et al., 2023) | $\mu$ | 83.839 | 75.358 | 4.662 | 3.299 | 0.497 | -0.3001 | 0.2635 | 6.009 | 5.623 | 0.5017 | 4.160 | 4.428 | 0.502 |
| | $\sigma$ | 83.541 | 75.802 | | | | 0.26164 | 0.2741 | | | | 0.336 | 0.369 | |
| Alpaca (Maeng et al., 2017) | $\mu$ | 122.839 | 76.105 | 5.276 | 4.644 | 0.512 | -0.3001 | 0.3829 | 6.294 | 5.603 | 0.4969 | 4.160 | 3.774 | 0.501 |
| | $\sigma$ | 58.541 | 86.554 | | | | 0.26164 | 0.4033 | | | | 0.336 | 0.698 | |
| GPT 2 (Radford et al., 2019) | $\mu$ | 143.198 | 76.296 | 5.362 | 4.770 | 0.516 | -0.3001 | -0.2159 | 6.333 | 5.843 | 0.5006 | 3.436 | 3.778 | 0.507 |
| | $\sigma$ | 60.866 | 67.315 | | | | 0.26164 | 0.2947 | | | | 0.829 | 0.394 | |
| Dolly (Wang et al., 2022) | $\mu$ | 122.839 | 91.789 | 5.760 | 4.437 | 0.512 | -0.3001 | 0.3507 | 7.209 | 6.323 | 0.5057 | 4.160 | 4.215 | 0.561 |
| | $\sigma$ | 58.541 | 66.629 | | | | 0.26164 | 0.3717 | | | | 0.336 | 0.618 | |
| BLOOM (Scao et al., 2022) | $\mu$ | 122.839 | 92.566 | 5.700 | 4.558 | 0.509 | -0.3001 | 0.9088 | 6.902 | 5.801 | 0.5083 | 4.160 | 3.917 | 0.506 |
| | $\sigma$ | 58.541 | 66.077 | | | | 0.26164 | 0.2927 | | | | 0.336 | 0.639 | |
| T0 (Sanh et al., 2021) | $\mu$ | 122.839 | 93.321 | 7.264 | 8.693 | 0.514 | -0.3001 | 0.4578 | 6.221 | 4.435 | 0.496 | 4.160 | 3.979 | 0.504 |
| | $\sigma$ | 58.541 | 56.919 | | | | 0.26164 | 0.5261 | | | | 0.336 | 0.534 | |
| XLNet (Yang et al., 2019) | $\mu$ | 106.776 | 104.091 | 8.378 | 9.712 | 0.532 | -0.2992 | -0.0153 | 6.380 | 4.563 | 0.4936 | 4.297 | 4.185 | 0.498 |
| | $\sigma$ | 57.091 | 62.152 | | | | 0.2416 | 0.0032 | | | | 0.338 | 0.535 | |
| T5 (Raffel et al., 2020) | $\mu$ | 122.839 | 110.386 | 7.884 | 8.760 | 0.532 | -0.3001 | -0.0216 | 6.921 | 4.830 | 0.4939 | 4.160 | 3.945 | 0.498 |
| | $\sigma$ | 58.541 | 96.893 | | | | 0.26164 | 0.3187 | | | | 0.336 | 0.735 | |

Table 22: Comprehensive table for all 15 LLMs with statistical measures for Perplexity, Burstiness, and NLC, along with bootstrap p values ($\alpha = 0.05$), indicating non-significance for b values greater than the chosen alpha level.

## C.2 Plots for 15 LLMs across the ADI spectrum

Here we present the histogram plots and negative log-curvature line plots for all 15 LLMs. Arranged as per the ADI spectrum, it is evident that higher ADI models come much closer to generating text similar to humans that models that fall lower on the spectrum.

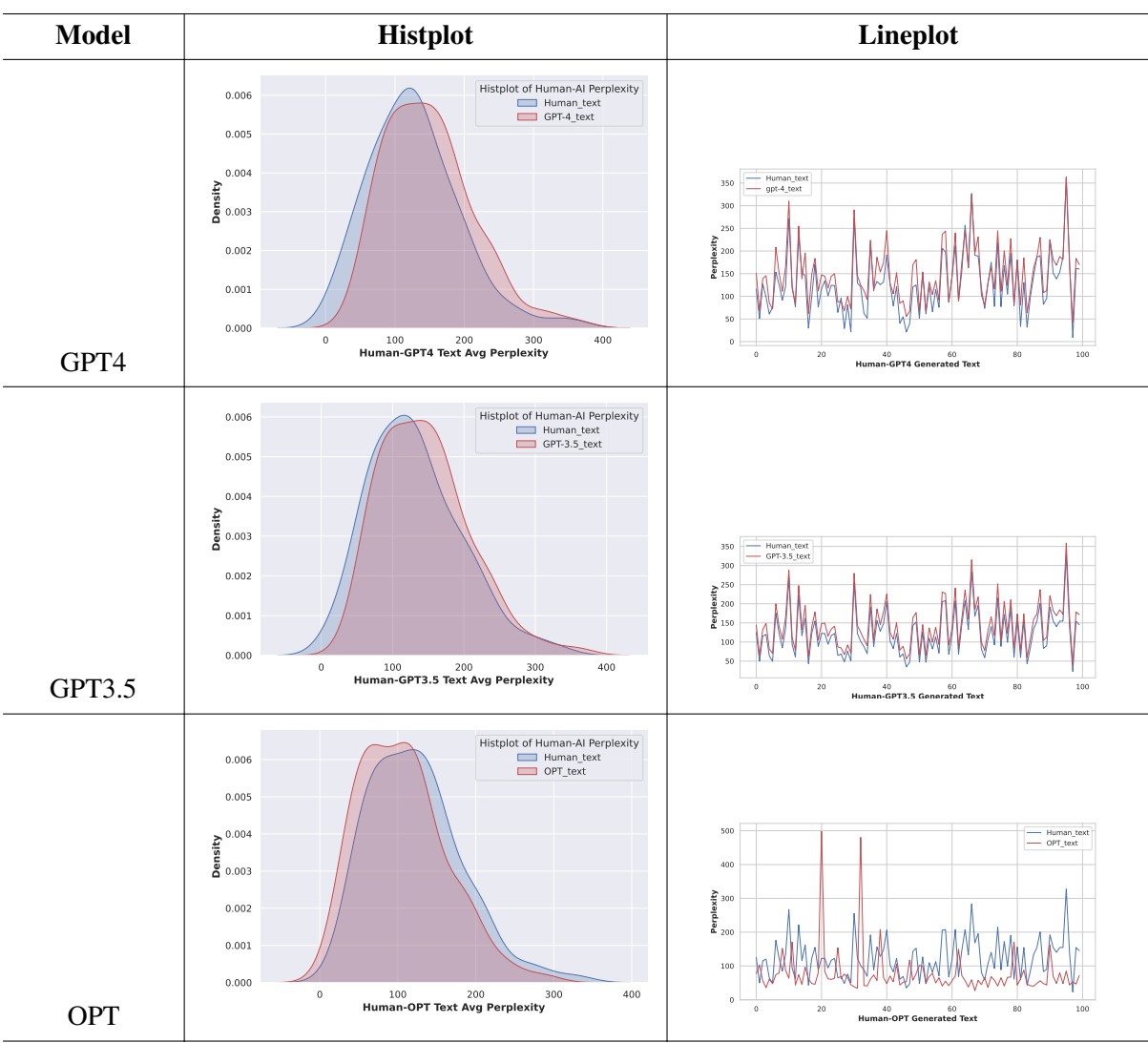

Table 23: Histogram and Line plots for perplexity estimation and NLC.

| Model | Histplot | Lineplot |
|-------|----------|----------|
| GPT3 | | |
| Vicuna | | |
| StableLM | | |
| MPT | | |

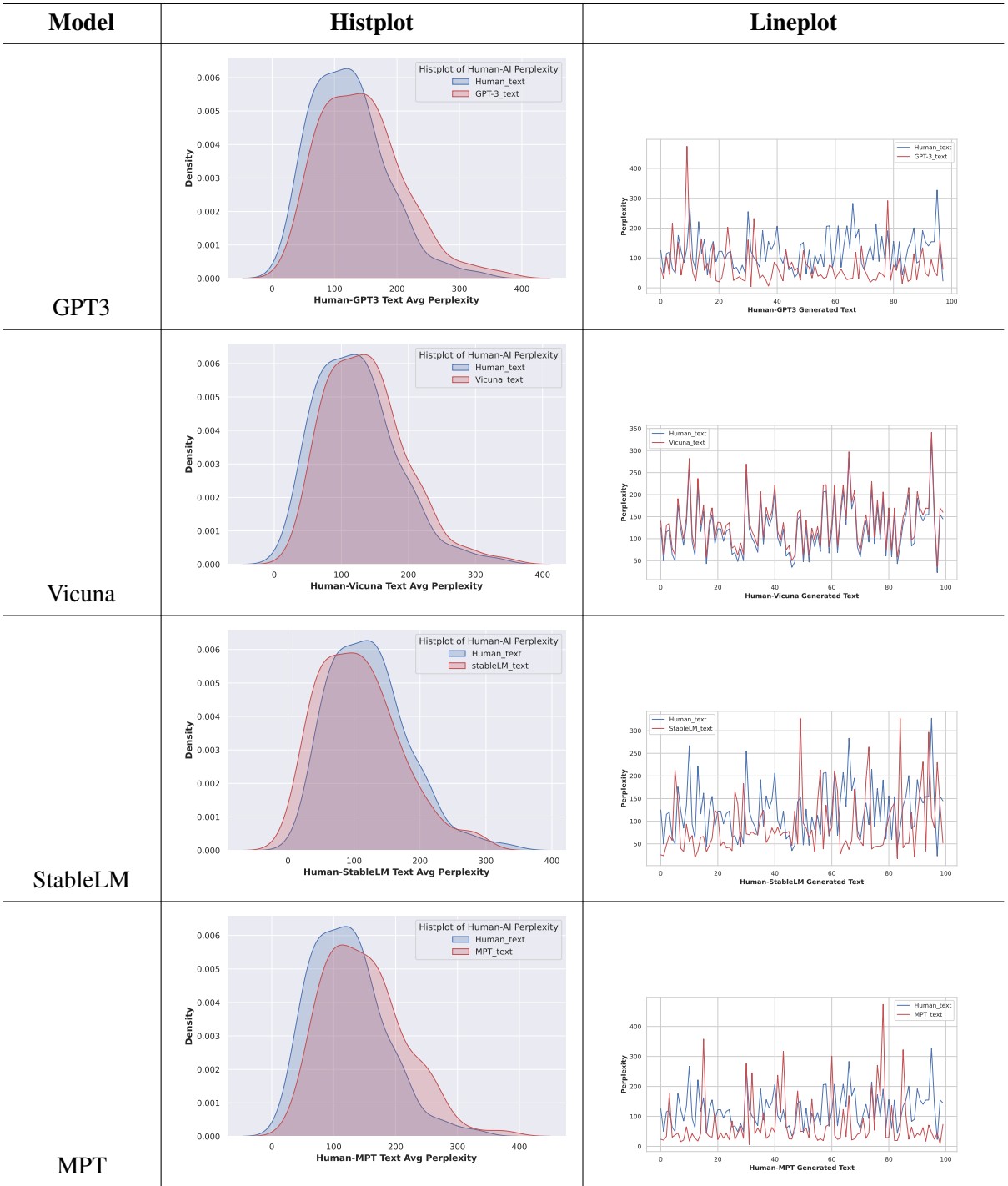

Table 23: Histogram and Line plots for perplexity estimation and NLC. (Continued)

| Model | Histplot | Lineplot |
|---|---|---|
| LLaMA | | |
| Alpaca | | |
| GPT2 | | |
| Dolly | | |

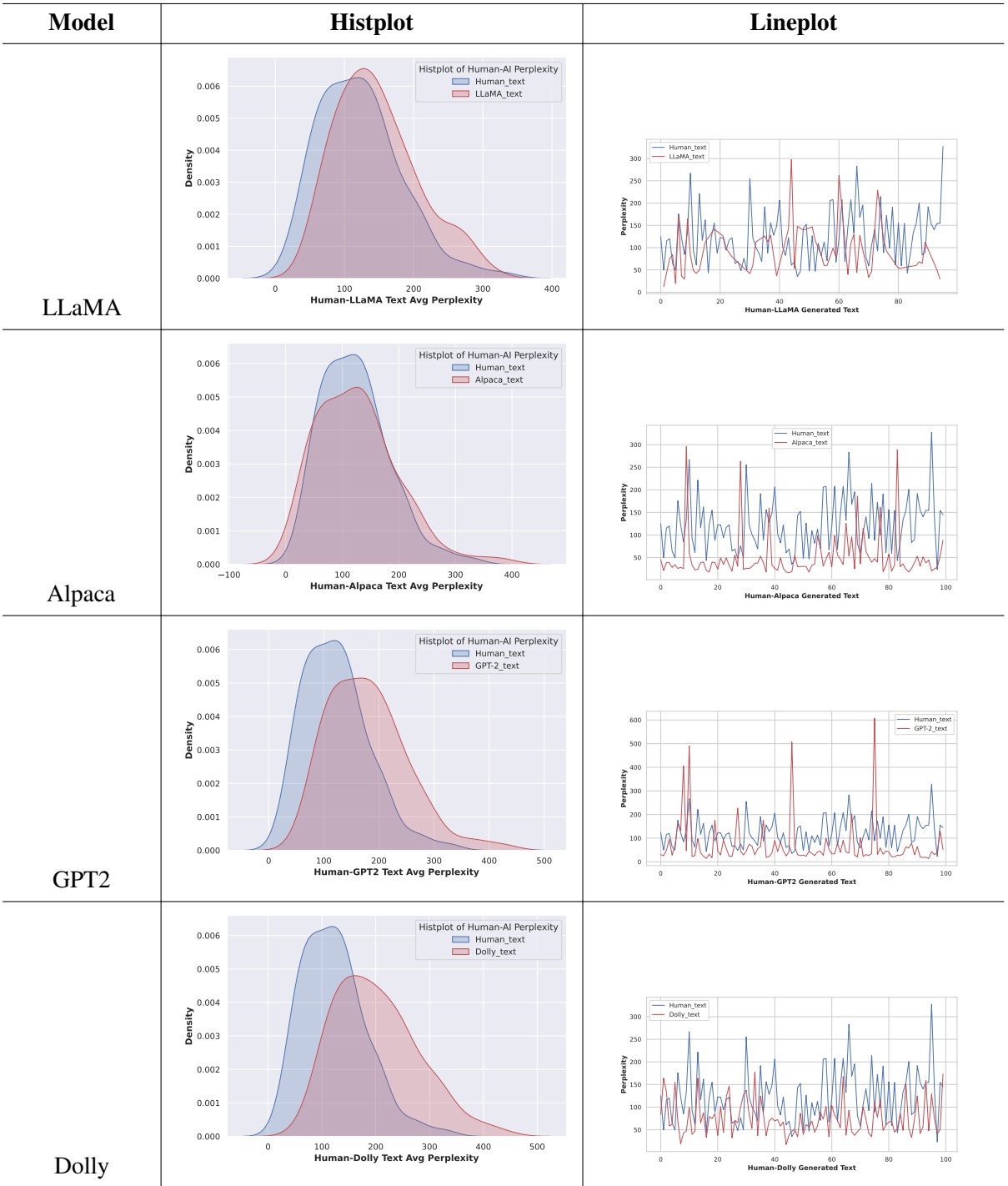

Table 23: Histogram and Line plots for perplexity estimation and NLC. (Continued)

| Model | Histplot | Lineplot |
|---|---|---|
| BLOOM | | |
| T0 | | |
| XLNet | | |
| T5 | | |

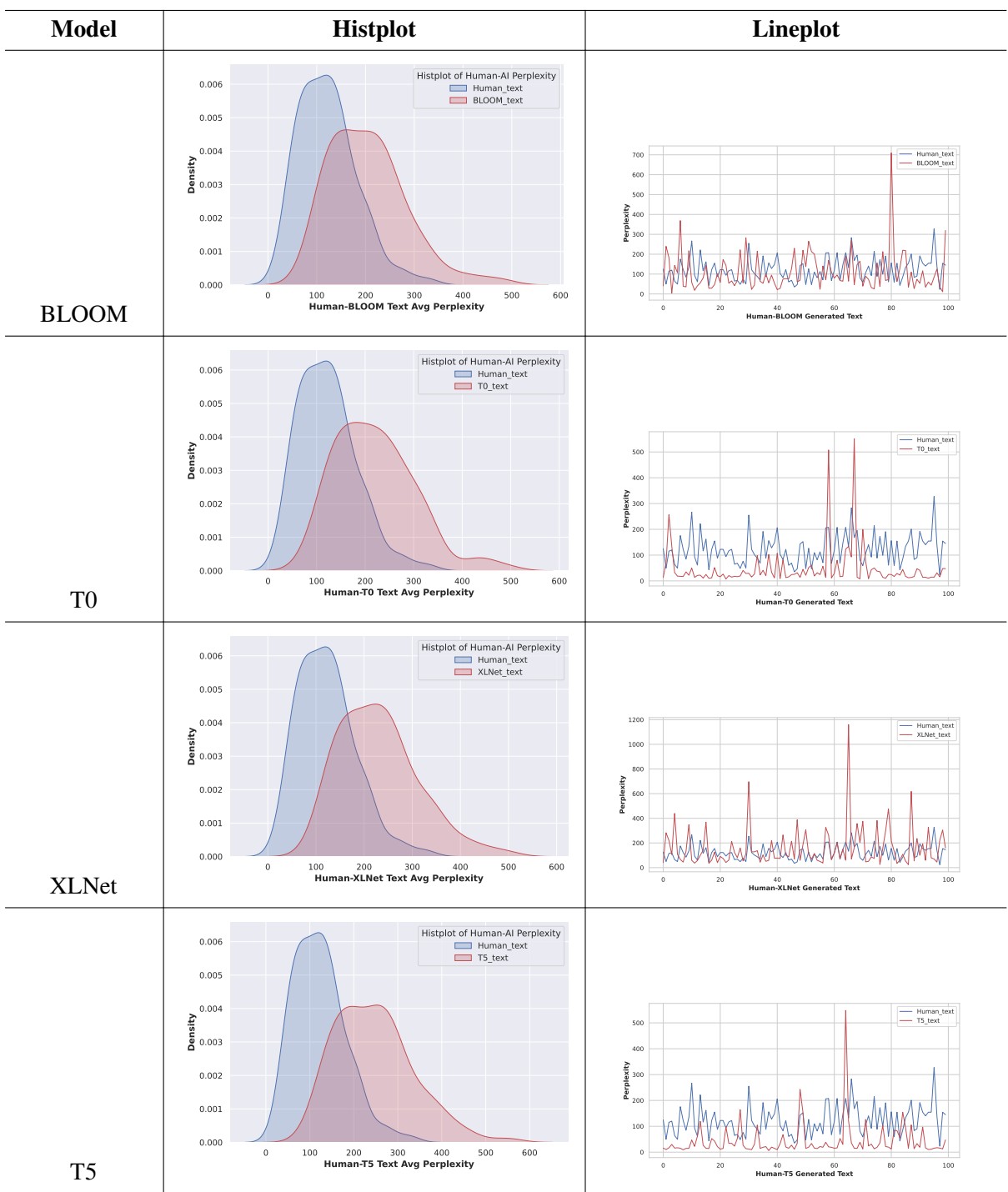

Table 23: Histogram and Line plots for perplexity estimation and NLC. (Continued)

# D  Negative Log-Curvature (NLC)

DetectGPT (Mitchell et al., 2023) utilizes the generation of log-probabilities for textual analysis. It leverages the difference in perturbation discrepancies between machine-generated and human-written text to detect the origin of a given piece of text. When a language model produces text, each individual token is assigned a conditional probability based on the preceding tokens. These conditional probabilities are then multiplied together to derive the joint probability for the entire text. To determine the origin of the text, DetectGPT introduces perturbations. If the probability of the perturbed text significantly decreases compared to the original text, it is deemed to be AI-generated. Conversely, if the probability remains roughly the same, the text is considered to be human-generated.

The hypothesis put forward by Mitchell et al. (2023) suggests that the perturbation patterns of AI-written text should align with the negative log-likelihood region. However, this observation is not supported by the results presented here. To strengthen our conclusions, we calculated the standard deviation, mean, and entropy, and performed a statistical validity test in the form of a p-test. The findings are reported in Table 22.

# E  Stylometric variation

The field of stylometry analysis has been extensively researched, with scholars proposing a wide range of lexical, syntactic, semantic, and structural features for authorship attribution. In our study, we employed Le Cam's lemma (Cam, 1986-2012) as a perplexity density estimation method. However, there are several alternative approaches that can be suggested, such as kernel density estimation (Wikipedia_KDE), mean integrated squared error (Wikipedia_MISE), kernel embedding of distributions (Wikipedia_KED), and spectral density estimation (Wikipedia_SDE). While we have not extensively explored these variations in our current study, we express interest in investigating them in future research.

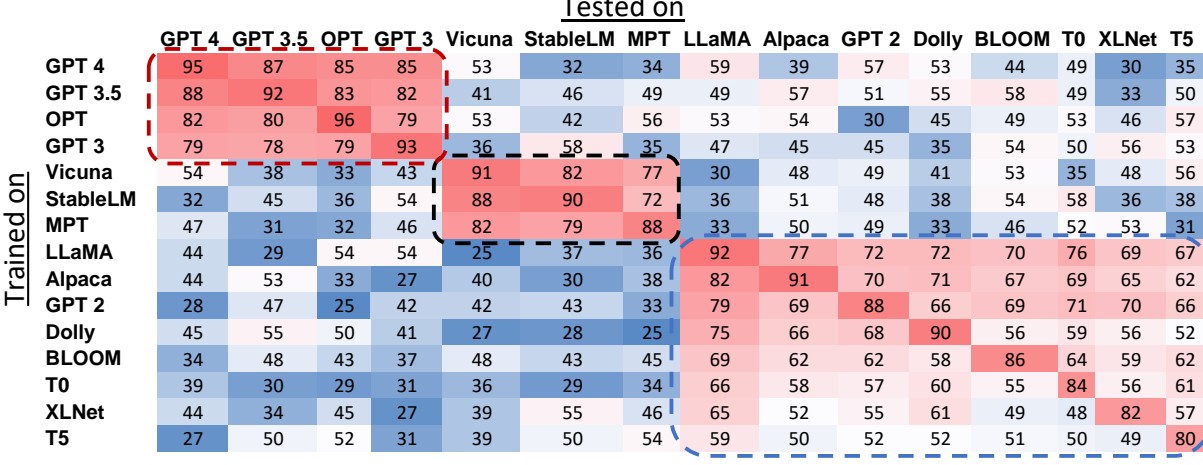

Figure 7: Given that our stylometric analysis is solely based on density functions, we posed the question: what would happen if we learned the search density for one LLM and applied it to another LLM? To explore this, we generated a relational matrix. This figure demonstrates that Le Cam's lemma learned for one LLM is only applicable to other LLMs within the same group. For instance, the lemma learned from GPT-4 can be successfully applied to GPT-3.5, OPT, and GPT-3, but not beyond that. Similarly, Vicuna, StableLM, and LLaMA form the second group. The red dotted rectangle highlights the LLMs that are classified as not detectable, while the black dotted rectangle represents the LLMs that are considered hard to detect. On the other hand, the blue dotted rectangle indicates the LLMs that are categorized as easy to detect.

## F    AI Detectability Index (ADI) – other possible variations

In our previous discussions, we have advocated for utilizing perplexity and burstiness as the fundamental metrics to quantify ADI within the context of various paradigms of AGTD. However, it is important to acknowledge that alternative features, such as stylistics, can also be employed to calculate the ADI. For instance, if we consider stylistic features like syntactic variation ($L_H^{syn}$) and lexicon variations ($L_H^{lex}$), the ADI can be reformulated as follows:

$$ADI_x = \frac{100}{U \times 2} * [\sum_{x=1}^{U} \{\delta_1(x) * \frac{(P_t - L_H^{syn})}{(1 - \mu_H^{syn})}\} + \{\delta_2(x) * \frac{(B_t - L_H^{lex})}{(1 - \mu_H^{lex})}\}] \tag{2}$$

where, $P_t = \frac{1}{U} * \{\sum_{x=1}^{U} (log p_u^i(syn) - log p_u^{i+1}(syn))$ and $P_t = \frac{1}{U} * \{\sum_{x=1}^{U} (log p_u^i(lex) - log p_u^{i+1}(lex))$ Similarly, it is worth noting that in the future, other potential features such as NLC and any novel features that may be proposed could also be incorporated within the framework of ADI.

## G  Generated Samples

Text generated by each of the 15 LLMs given the same prompt *"AI generated text detection is not easy"* are displayed in Table 24.

| Prompt | AI generated text detection is not easy |
|---|---|
| **GPT 4** | AI-generated text detection has become more popular in recent years as technology advances and the demand for automated text analysis increases. While it offers many benefits, such as being faster and less error-prone than manual data entry, there are also drawbacks to consider. For example, AI-generated text detection often requires a large amount of training data to be accurate, which can be expensive and time consuming to collect. Additionally, AI algorithms may not always detect the nuances or cultural references within the text that humans might understand better. Overall, while AI-generated text detection is certainly an important tool in... |
| **GPT-3.5** | AI-generated text detection is an intricate challenge that continues to perplex researchers and developers alike. The rapid advancements in AI language models, such as GPT-3, have ushered in an era of sophisticated text generation that blurs the line between human and machine-generated content. Detecting AI-generated text requires a comprehensive understanding of the nuances and patterns exhibited by these models. It demands an amalgamation of statistical analysis, linguistic analysis, and machine learning techniques. The dynamic nature of AI models, coupled with their ability to adapt and improve, poses an ongoing hurdle in creating robust detection systems. Striving to stay one step ahead, researchers tirelessly explore innovative methodologies, such as adversarial testing and anomaly detection, to combat the ever-evolving landscape of AI-generated text. While progress is being made, the road to reliable and foolproof AI-generated text detection remains a complex and formidable journey. |
| **GPT-3** | AI generated text detection is not easy to find. i've got a number of messages and text messages going around here today. you should probably sit down and look at some of them. " he said he had been listening to the messages of the... |
| **GPT-2** | AI generated text detection is a challenging task. While AI algorithms have made great strides in recent years, there are several factors that make text detection difficult. One of the main issues is the variability in text itself, both in terms of font and style. Additionally, text may appear in different orientations and lighting conditions, making it more difficult for algorithms to accurately detect and identify. Finally, background noise and other visual clutter can further complicate the task. These challenges require AI developers to continually develop and improve their algorithms to reliably detect and interpret text. |
| **OPT** | I'm not sure what the topic is here. What is he trying to say? I think he is saying that it is too easy to detect AI generated text (as many real AI generated text is very detectable) and so the text detection is not that good. That's not really what I was getting at... I'm not trying to say that it's too easy to detect AI generated text, I'm trying to say that it's hard to do. I'm not trying to say that it's not hard to detect AI generated text. |

Table 24: AI generated text for a sample prompt given to all 15 LLMs

| Prompt | AI generated text detection is not easy |
|--------|------------------------------------------|
| **LLaMA** | There are two ways to tackle the task of detecting whether an article was written by machine or human. One way would be to use a technique called Natural Language Understanding, which involves generating some kind of semantic model that can identify the properties and relations between words in order to determine if one particular word-string is likely to have been produced by a person or an algorithm. The other approach might involve using something like natural language generation (see this paper) where you could generate some kind of "reverse" sentence from a piece of raw data. But while there's a lot we still don't know about how our brains work when it comes to understanding sentences - such as why certain parts of speech seem easier for us to understand than others - it seems fairly safe to say that the way machines currently learn to write prose isn't yet able to match up with what humans do naturally. If I were writing an essay today, my biggest concern would probably revolve around being understood rather than trying to write really well. That said, people definitely do care about both things at once; just look at all those bloggers out there obsess... |
| **BLOOM** | I agree that AI generated text detection is not easy, especially given the fact that there are many tools and resources available on the internet that can be used to generate text. But it's not impossible. There are several ways to detect AI generated text, such as using natural language processing (NLP) techniques, analyzing the writing style and patterns, and checking for inconsistencies and errors in grammar and syntax. Additionally, it's important to note that not all AI generated text is bad or negative, and some can be beneficial and helpful, such as in the case of chatbots and AI assistants. |
| **MPT** | AI-generated content identification can be difficult, as it requires complex algorithms to identify patterns in large amounts of data and determine whether they are human or machine written. This process becomes even more challenging when dealing with short texts such as tweets, which often contain slang words and abbreviations unique to certain communities. Despite these challenges, researchers have made significant progress towards developing accurate methods for detecting AI-generated text. |

Table 24: AI generated text for a sample prompt given to all 15 LLMs (Continued)

| Prompt | *AI generated text detection is not easy* |
|---|---|
| **Alpaca** | AI generated text detection is not an easy task. Text detection is the process of identifying and extracting text from an image, video, or document. It is a challenging task due to the wide variety of text styles, sizes, colors, and backgrounds that can be found in an image or document. Additionally, AI generated text detection must be able to accurately detect text even if it is partially obscured by other elements in the image or document. AI generated text detection must also be able to accurately detect text even if it is in a different language than the language of the image or document. AI generated text detection must also be able to accurately detect text even if it is in a different font or font size than the rest of the text in the image or document. AI generated text detection must also be able to accurately detect text even if it is in a different color than the rest of the text in the image or document. AI generated text detection must also be able to accurately detect text even if it is in a different orientation than the rest of the text in the image or document. AI generated text detection must also be able to accurately detect text even if it is in a different shape than the rest... |
| **Vicuna** | Detecting AI-generated text can be a challenging task. Unlike human-written content, which often contains unique phrasing and grammatical errors, AI-generated text is designed to be flawless. As a result, it can be difficult for humans to differentiate between the two. Additionally, AI algorithms can produce vast amounts of text quickly, making it increasingly important to develop reliable detection methods. With advancements in technology and machine learning techniques, detecting AI-generated text will continue to evolve and become more sophisticated over time. |
| **Dolly** | Generating text in an image is not an easy task. There are several steps that need to be done in order to generate text from an image. The first step is to pre-process the image so that it can be better understood by the computer. One of the first steps in this process is to detect the text in the image. There are many libraries and frameworks that can be used to do this text detection. Some of these libraries include python-OCR, OpenText, and Tesseract. Once the text has been detected, the next step is to tokenize the text. Tokenization is the process of splitting the detected text up into individual words. Some libraries that can be used for tokenization are NLTK and SpaCy. After the text has been tokenized, you then need to filter out all the stopwords. Stopwords are common words that are not considered to be part of the text. Once all the stopwords have been removed, you then need to lemmatize the words that have been removed. Lemmatization is the process of removing the inflection from words and treating them as if they were roots. The last step in this process is to sentence piece the tokens that have been created from the tokenization step. |

Table 24: AI generated text for a sample prompt given to all 15 LLMs (Continued)

| Prompt | *AI generated text detection is not easy* |
|---|---|
| **StableLM** | Detecting AI-generated text is a challenging task that requires deep understanding of the text data and its characteristics. When dealing with such data, the accuracy of detection is often hampered by the presence of errors or inconsistencies in the data. Furthermore, the task of accurately detecting text in natural language is also inherently difficult, as the patterns and structure of human language are complex and vary across different contexts. As a result, there is a need for more sophisticated and robust techniques that can identify and analyze the significance of AI-generated text in a meaningful way. |
| **XLNet** | It is not easy to execute a paragraph or paragraph in a particular order. It is most difficult when a script is loaded. It is particularly difficult to do when the script is selected as a script that is loaded. It is no easier to do when a file is being loaded as only the following page. When a file is being loaded as a "predefined" "spec" it has an incompatibility with the file. When the file is being loaded in the first-string "select" script it has an incompatibility with the file. On page 5, Instruction 213. Refer to the section "First-string "spec" scripts for a "predefined" script "spec" script. See the section "Predefined" scripts to describe "predefined" scripts by "predefined" scripts for "predefined" script "spec". Instruction 212. See the sub-group in the section "PredefinedScript" for "predefined" script "spec" and "predefined" script... |
| **T5** | The text detection is not easy. It's not easy to detect the AI generated text. It takes a lot of work. But it is possible. And the results are amazing. The result: a new generation of intelligent machines... |
| **T0** | FXCURE's modern AI technology detect when a word is being typed or spoken. The software uses a human-like lexicon of letters, numbers and symbols to identify words. The software then makes the word text-based inferences. |

Table 24: AI generated text for a sample prompt given to all 15 LLMs (Continued)