# OpenReview forum: "Counter Turing Test (CT2): AI-Generated Text Detection is Not as Easy as You May Think - Introducing AI Detectability Index (ADI)"
_EMNLP/2023/Conference — EMNLP 2023 Main_

### Official Review · Reviewer_6tjm · 2023-08-04

**Soundness:** 4

**Excitement:**

4: Strong: This paper deepens the understanding of some phenomenon or lowers the barriers to an existing research direction.

**Paper Topic And Main Contributions:**

The paper experimentally compares a number of AI-Generated Text Detection techniques applied to a variety of Large Language Models. They show that these detection techniques are not effective at detecting text generated by state-of-the-art Large Language Models. The authors then introduce the AI Detectability Index which, based on a metric that combines some AGTD techniques, ranks LLMs on a scale of how easily text generated by that particular LLM can be identified as being AI-generated.

**Reasons To Accept:**

There is much discussion in the public sphere and in the literature about AI Generated Text, and AGTD techniques are urgently sought after. This paper contributes meaningfully to that discussion by their robust empirical analysis of existing AGTD techniques.

**Reasons To Reject:**

The presentation of the paper is a mess.

**Reproducibility:**

4: Could mostly reproduce the results, but there may be some variation because of sample variance or minor variations in their interpretation of the protocol or method.

**Reviewer Confidence:**

3: Pretty sure, but there's a chance I missed something. Although I have a good feel for this area in general, I did not carefully check the paper's details, e.g., the math, experimental design, or novelty.

---

> ### Author Rebuttal · Authors · 2023-08-28
>
> Thanks for your review, however we are discouraged by your scores. We are addressing your concerns in detail hoping to receive better scores.
>
>
> Reasons to reject:
>
> ---------------------------
>
> “The proposed ADI just combines some existing metric for AGTD, without too much novlty.” -
>
> Our definition of ADI is a composition of two linguistic measures: lexical measure (burstiness) and syntactic measure (perplexity). We have combined them based on empirical observation using a density function defined by Le Cam’s Lemma. We have also discussed and self-criticized in Appendix F, pg. 28/33, that ADI could be reformulated using similar, alternative features. Finally, we have discussed and hinted how future researchers may extend the definition of ADI.
>
> Additionally, regarding the novelty of the work, to the best of our knowledge, such a metric or definition has not yet been formally pursued in our community of researchers, or on this topic. If you disagree, we would really appreciate you citing resources for the same.
>
> “The presentation of the paper is a bit mess”-
>
> Given the extensive depth of our work, presenting it efficiently under the constraints of format and page limitations was indeed a herculean task. We appreciate all your feedback on the presentation of our work. As we are reworking for the camera-ready version of the paper, we will certainly consider these remarks. If this paper gets accepted, we will dedicate efforts to improving the presentation with the extra granted page for adding content.
>
> ---------------------------
>
> “Excitement: 2: Mediocre: This paper makes marginal contributions (vs non-contemporaneous work), so I would rather not see it in the conference.”
>
>
> Given the recent spike of interest in Generative AI, we believe that this work is a timely contribution in pointing out the current ineffectiveness of detection techniques. Furthermore, discussions on AI regulations and policies are gaining serious attention from AI enthusiasts. Our contribution of ADI to aid policy making is also a major contribution of this work. If you can please provide specific criticism justifying your scores, we would happily address them.
>
>
> The other two reviewers have given good scores reflecting the technical merit of this work. We note your concern on the presentation and will address any specific guidance to improve it. However, we hope your scores reflect the technical merit including the novelty of this work.

---

### Official Review · Reviewer_weMu · 2023-08-05

**Soundness:** 5

**Excitement:**

5: Transformative: This paper is likely to change its subfield or computational linguistics broadly. It should be considered for a best paper award. This paper changes the current understanding of some phenomenon, shows a widely held practice to be erroneous in someway, enables a promising direction of research for a (broad or narrow) topic, or creates an exciting new technique.

**Paper Topic And Main Contributions:**

The paper investigates an extremely important issue of identifying model generated text. The authors introduce a Counter Turing Test evaluating the robustness of different methods to identify AI-generated texts. The authors experiment with 15 different large language models and a human written text showing that different AI-detection methods are not robust enough to server their purpose. The experiments are valuable and appear to be relatively well designed. Finally, the authors propose an AI detectability index, which will allow to rank the models according to their detectability.

**Questions For The Authors:**

Question A: Do you think your findings will generalize to other languages? (e.g., the same methods work/don't work, LLMs are ultimately not detectable by the current set of methods (or rather by one method alone))

Question B: In terms of identifying the style, wouldn't it be different than a style of human writer. The bigger language models are very capable of producing texts in tailored styles.

**Reasons To Accept:**

- The paper addresses a very important issue introducing a benchmark to diagnose the robustness of different methods to identify an AI-generated text.

- The authors experiment with 15 different large language models testing the performance of different methods in identifying AI-generated text in English. The analysis is throughout and the authors clearly did a great job reviewing all the existing methods. The introduced benchmark is very much needed in the field right now.

- The paper writing is excellent, with many issues discussed and being made very up to date.

- The choice of New York Times is well justified and appropriate for the experiments. The statistical analysis appears to be sound.

**Reasons To Reject:**

I do not think this paper should be rejected, but here are my two main concerns:

- The limitation section added at the end of the paper (beyond page 8) is very good but also extremely long. This would not be an issue, if not for the fact that it seems to cover more than the limitations. I really wish some of this was either in the main body or in a proper section of the appendix so I did not have to make this statement.

- The conclusions are English-exclusive but the authors do not discuss or mention it anywhere (actually I don't think the language is mentioned anywhere throughout the paper).

**Reproducibility:**

4: Could mostly reproduce the results, but there may be some variation because of sample variance or minor variations in their interpretation of the protocol or method.

**Reviewer Confidence:**

4: Quite sure. I tried to check the important points carefully. It's unlikely, though conceivable, that I missed something that should affect my ratings.

**Typos Grammar Style And Presentation Improvements:**

Please state clearly that the work was done for and on English. While this work is using also multilingual language models the performance of different methods is tested for English only.

Lines 1076-1091 are misleading. DIPPER was trained and first introduced in Krishna et al. (2023). It was later added by Sadasivan et al. (2023) in a paper update.


I might have missed this, but if it just wasn't there, I believe that a discussion about bad actors exploring the ADI is needed.

Please check your references for proper formatting (capital letters -- e.g., "eu" -> "EU") and duplicates ("Normalization" wiki entry), personally, I would try citing a source different from Wikipedia, but nowadays this may be more a matter of preferences rather than correctness.

---

> ### Author Rebuttal · Authors · 2023-08-28
>
> Thanks for your excellent review and questions, find out detailed response below -
>
> Reasons to reject:
>
> ----------------------------
>
> “limitation section”
>
> We were unable to include some key points in the main section due to the page limitations. However, we appreciate your feedback that “it seems to cover more than the limitations” and will certainly address this in the final version of the paper.
>
> “English-exclusive but the authors do not discuss or mention it anywhere”
>
> Thank you for pointing this out. Since this paper reports experiments only for the English language, we will specify this in the abstract.
>
>
>
> Questions to authors:
>
> ---------------------------------
>
> Question A: “generalize to other languages…”
>
> Answer A: We have designed all the experiments and ADI in a way that is fairly applicable to any language. For example, de-watermarking techniques that we have discussed are based on “entropy" calculation, which is language agnostic. We have defined ADI primarily based on perplexity and burstiness, which could be applied for any language. Additionally, to expand the scope of our claim, we are already working on other languages, such as Spanish and Hindi, which we hope to publish soon.
>
> Question B: “In terms of identifying the style, wouldn't it be different…”
>
> Answer B: This is a great point. We are aware that models like GPT4, LLaMA can mimic various styles if explicitly instructed to do so. However, in this paper, we have only applied stylometric study on LLM's default linguistic style. Interestingly, while identifying style through stylometric analysis, which in our work is based on density functions of perplexity and burstiness, we have indeed observed different findings in LLMs than that of human authors. However, the key finding, as revealed in fig. 7 (pg. 29/33), is that LLMs can be grouped based on their linguistic styles, which provides us with insightful features in possibly detecting AI-generated text.
>
> Typos Grammar Style And Presentation Improvements:
>
> ---------------------------------------------------------------------------------------
>
> All points are well taken and will be addressed in the final camera-ready version of the paper.
>
>
> “DIPPER was trained and first introduced in Krishna et al. (2023). It was later added by Sadasivan et al. (2023)” - Thanks for pointing out this overlook. We will fix it.
>
>
> “discussion about bad actors exploring the ADI is needed.” - Yes, we have said so in the Ethical Considerations section on page 11, line #742
>
>
> “Please check your references” - We will fix errata in the reference. Appreciate your suggestions.
>
>
> Finally, we urge you to consider upgrading the excitement score, which allows us to compete for the best paper award in the track.

---

### Official Review · Reviewer_WEM6 · 2023-08-05

**Soundness:** 5

**Excitement:**

5: Transformative: This paper is likely to change its subfield or computational linguistics broadly. It should be considered for a best paper award. This paper changes the current understanding of some phenomenon, shows a widely held practice to be erroneous in someway, enables a promising direction of research for a (broad or narrow) topic, or creates an exciting new technique.

**Paper Topic And Main Contributions:**

The paper presents a highly contemporary yet pernicious problem of detecting AI-generated text reliably and provides empirical evidence on the shortcomings of pre-existing approaches. Their experiments span six methods for detecting AGTD over 15 contemporary LLMs. To reliably test their findings, they leverage a massive dataset of human and machine-generated text (CT^2). Their results reveal the brittleness of current methods and how they are easy to circumvent, primarily as models scale. Subsequently, the work introduces the notion of AI Detectability Index for LLMs to infer whether their generations can be detected automatically.



**Questions For The Authors:**

It seems a bit counter-intuitive to name the metric as AI Detectability Index, since a higher score on the index would signal that the model's text is easier to detect.

**Reasons To Accept:**

1. A timely paper identifying the brittleness of pre-existing techniques for detecting AI-generated text.
2. They create an open-sourced benchmark dataset (CT^2)  and introduce the AI detectability index (ADI) as a means for researchers to assess whether the text generated by LLMs is detectable or not.
3. The paper is well-worded and provides an in-depth view of contemporary AGTD methods.
4. The paper has significant implications for shaping future policies and regulations related to AI and thus holds immense importance to the NLP community.




**Reasons To Reject:**

None

**Reproducibility:**

5: Could easily reproduce the results.

**Reviewer Confidence:**

3: Pretty sure, but there's a chance I missed something. Although I have a good feel for this area in general, I did not carefully check the paper's details, e.g., the math, experimental design, or novelty.

---

> ### Author Rebuttal · Authors · 2023-08-28
>
> Thank you for your time and reviews; we are glad you like this work.
>
> Questions to authors:
>
> ---------------------------------
>
> Question: “counter-intuitive to name the metric as AI Detectability Index”
>
> Response: Agreed, this has also been a topic of discussion and debate among us authors. A straight-forward way of addressing this is to use an inverse function in the ADI formulization.

---

### Meta-Review · Area_Chair_v3th · 2023-09-22

**Recommendation:** 5

**Metareview:**

This study introduces a dataset and a method for analyzing generated text. The research delves into existing methods and identifies their limitations. Subsequently, the proposed method is evaluated on 15 LLMs.

All reviewers acknowledged the paper's timely contribution, which promises significant benefits for the community. Reviewer WEM6 expressed a concern about the name "AI Detectability Index", this could be taken into account while improving the paper.

Reviewer weMu emphasized the importance of understanding the method's generalizability to other languages. Including a discussion on this aspect would greatly benefit readers.

---

### Decision · Program_Chairs · 2023-10-07

**Decision:**

Accept-Main

**Comment:**

This study introduces a dataset and a method for analyzing generated text. The research delves into existing methods and identifies their limitations. Subsequently, the proposed method is evaluated on 15 LLMs.

All reviewers acknowledged the paper's timely contribution, which promises significant benefits for the community. Reviewer WEM6 expressed a concern about the name "AI Detectability Index", this could be taken into account while improving the paper.

Reviewer weMu emphasized the importance of understanding the method's generalizability to other languages. Including a discussion on this aspect would greatly benefit readers.